# Extracellular matrices regulate extravasation journey of leukocytes and inflammatory tissue fate

**Yu-Tung Li***

Department of Stem Cell Therapy Science, Graduate School of Medicine, The University of Osaka, Suita, Japan

## eLife Assessment

This Review Article provides a timely review of how the extracellular matrix (ECM), particularly the vascular basement membrane, regulates leukocyte extravasation, migration, and downstream immune function, with a focus on monocytes/macrophages. It integrates molecular, mechanical, and spatial aspects of ECM biology in the context of inflammation, drawing from recent advances.

**Abstract** Leukocyte extravasation across the blood endothelium to inflamed tissues is a crucial defence mechanism against invading pathogens. After the elimination of the pathogen in the tissue, inflammation needs to be resolved back to steady state. This cascade comprises at least three stages: transmigration through the endothelium and the underlying basement membrane, intra-tissue leukocyte activity, and tissue resolution. In each stage, extracellular matrix proteins in the vascular basement membrane and in tissues regulate a multitude of endothelial and leukocyte functions essential to completion of the cascade, either as a collective force-permissive structure or through signaling by individual matrix proteins. Proper orchestration of these diverse processes during the extravasation journey ensures effective defence and avoids development of chronic inflammatory diseases. This review will focus on how these extracellular matrices regulate the extravasation journey of leukocytes to illustrate their tight functional interdependence with profound impacts on the ultimate post-inflammation tissue fate.

**\*For correspondence:**
tomli_yt@sts.med.osaka-u.ac.jp

**Competing interest:** The author declares that no competing interests exist.

## Introduction

Cells in an inflamed tissue secrete cytokines to activate local blood vessels and command leukocyte infiltration. Tissue entry of all kinds of leukocytes must be optimally controlled such that the cause of the inflammation, potentially an invading and readily proliferative pathogen, can be timely eliminated. Latency of tissue entry depends on the leukocyte types, typically with neutrophils as the quickest responders (in hours), followed by monocytes (in hours to days) and lymphocytes (in days). Despite the different latency, they often follow common principles for tissue entry. Since the immune attack often involves cytotoxic substances such as granzymes and proteases, an overdose of leukocyte entry would cause an acute shock and tissue damage. Instead of a passive barrier for leukocytes to penetrate, the endothelium actively controls the leukocyte traffic via expression modulation of apical adhesion or surrogate molecules and junctional remodeling (*Nourshargh and Alon, 2014*; *Vestweber, 2015*). Only specific sites of the blood vessels, with moderate to low shearing blood flow and optimal expression of supportive endothelial molecules, are permissive to leukocyte exit from the circulation (extravasation). These extravasation sites exist primarily in post-capillary venules, venules, or specialized blood vessels such as the high endothelial venules (HEV) and the liver sinusoids (*Vestweber,*

*2015*; *Bogoslowski et al., 2018*; *McNamara and Cockburn, 2016*). Understanding regulation details of these extravasation events is fundamental to devise treatments for various inflammatory diseases.

Along the extravasation journey to fight pathogens, leukocytes are prone to functional modulation, solely through interaction with their surroundings. The resultant leukocyte activities determine whether pathogens can be cleared, and if so, whether the inflamed tissue can be restored to homeostasis. Macrophages derived from either tissue residence or extravasated monocytes play a pivotal role in this latter resolution process.

Across this multi-stage paradigm, extracellular matrix (ECM) proteins in the vascular basement membrane and in tissue play significant regulatory roles. Overviews of the leukocyte extravasation processes and intra-tissue leukocyte activities regulated by these ECM are given in Figures 1–2 and 3, respectively. This review focuses on the regulatory roles of ECM along the extravasation journey of leukocytes to inflamed tissue. This journey has intimate involvement with endothelial cells (EC), especially with their junctional remodeling events. The processes regulated by ECM discussed in this review are summarized in *Table 1*. Following chronological order of the extravasation journey, we will describe what vascular structures constitute extravasation barriers for leukocytes ('Venular architecture and barriers for leukocyte extravasation', 'Paracellular versus transcellular route of entry across the endothelial barrier'), how the leukocytes overcome these barriers ('Vascular basement membrane supports force-dependent endothelial junctional remodelling required for leukocyte diapedesis across the endothelium', 'Pre-existing barrier weak spots with low laminin-α5 expression guide leukocyte diapedesis'), how crossing these barriers affects leukocyte functions ('Barrier passages modify leukocyte functions in tissue via biophysical stress, proteolysis, selective leukocyte passage and spatial access to functional regulators', 'How do leukocytes overcome barriers without vascular leak?'), and how ECM interaction with leukocyte activities in tissue determines post-inflammatory tissue state ('Leukocyte proteases fragment tissue ECM to generate matrikines which promote secondary leukocyte trafficking', 'Tissue stiffness provides biophysical cues to modify extravasated leukocyte activities', 'Macrophage efferocytosis regulated by ECM is critical to timely subside inflammation and restore tissue homeostasis'). At the end of this review, we will also discuss why control of macrophage cell state may hold the key for effective modulation of inflamed tissue state to revive tissue homeostasis and how different macrophage cell states regulate ECM functions ('Macrophage efferocytosis regulated by ECM is critical to timely subside inflammation and restore tissue homeostasis', 'Prospects in inflammatory tissue fate modulation via navigating macrophage cell states').

## Venular architecture and barriers for leukocyte extravasation

In a blood venule, cells and extracellular components are organized in a defined manner to form extravasation barriers for leukocytes. The most inner circumference of the venule lines a monolayer of EC directly exposed to the shearing blood flow. Except for fenestrated blood vessels, this monolayer is typically continual, with individual ECs intercalated via adherens junctions localized at the intercellular basolateral side of the plasma membrane. The endothelial adherens junctions contain various adhesion molecules, such as vascular endothelial cadherin (VE-cadherin), junctional adhesion molecules (JAMs), PECAM-1, and CD99 (*Vestweber, 2015*). In peripheral EC, tight junction molecules such as claudin-5 and ZO-1 are expressed in lower quantities, whereas strong expression of these molecules is detected in EC of some specialized tissues such as the blood-brain barrier and the blood-retinal barrier in brain and retina, respectively. These junctional proteins are not rigidly held in place but under constant balancing actions of endocytosis and exocytosis of junctional proteins such that dynamic remodeling of the junctional tightness is possible. This, in turn, affects leukocyte traffic and vascular permeability. The most abluminal side of the venule is decorated with scattered perivascular cells (pericytes; *Figure 1A*). Lacking a definitive pan-pericyte marker, pericytes are defined as mesenchymal cells at vascular proximity, with subsets expressing markers of PDGFRβ, CD146, NG2, and/or α-SMA (*Hirschi and D'Amore, 1996*).

In between the EC monolayer and the porous pericyte layer is an ultra-thin adhesive layer of basement membrane composed of a network of self-assembled ECM proteins (*Lash et al., 1989*; *Timpl and Brown, 1994*). Constituent ECM proteins mainly comprise laminins (α4 and α5 chains), type IV collagen, nidogens, and perlecan (*Timpl, 1996*; *Pozzi et al., 2017*). These components bind endothelial integrins organized in focal adhesions to allow EC adhesion (*Pozzi et al., 2017*; *Brown et al., 1997*) and support establishment of cell polarity and intercellular junctions to limit undesirable tissue

**Table 1.** Biological processes regulated by extracellular matrix proteins or structures at each stage along the leukocyte extravasation journey.

**Extravasation journey stage:**
**Paracellular trans-endothelial diapedesis**

| ECM interaction (protein or structure) | Function | Reference |
|---|---|---|
| Vascular basement membrane/endothelial focal adhesions | Sense shear stress in blood flow to secrete PDGFs for pericyte recruitment and Ang1/Tie2-mediated junctional strengthening. | *Hsieh et al., 1991*; *Lindblom et al., 2003*; *Teichert et al., 2017* |
| | Sense shear stress in blood flow to endocytose VE-PTP to activate Tie2-mediated junctional strengthening. | *Shirakura et al., 2023* |
| | Enable endothelial force generation pulling on VE-Cad/cateinins/actomyosin complex to expose Y731 site for dephosphorylation by SHP-2, allowing subsequent VE-Cad endocytosis and local junctional destabilization around the passaging leukocyte. | *Arif et al., 2021*; *Wessel et al., 2014* |
| | Enable kinesin-mediated delivery of LBRC along an anchored microtubule network to endothelial junction around the passaging leukocyte for junctional destabilization. | *Mamdouh et al., 2008*; *Mamdouh et al., 2003*; *Dalal et al., 2021* |
| Laminin-511/endothelial Integrin α6β1 | Activate RhoA/ROCK to promote junctional deposition of VE-Cad to stabilize junction. | *Song et al., 2017* |
| | Promote p120 association to stabilize junctional VE-Cad. | *Di Russo et al., 2017* |

Extravasation journey stage:
Trans-basement membrane diapedesis

| ECM interaction (protein or structure) | Function | Reference |
|---|---|---|
| Laminins / α6 integrins | Allow migration through the basement membrane barrier. | *Dangerfield et al., 2002*; *Wang et al., 2005* |
| Vascular basement membrane/ leukocyte proteases | Locally breach the basement membrane barrier. | *Wang et al., 2006*; *Wang et al., 2005*; *Voisin et al., 2009* |
| | Produce ECM carryovers on extravasated leukocytes. | *Wang et al., 2006*; *Voisin et al., 2009* |
| Laminin carryovers on neutrophils | (Unknown) | *Wang et al., 2006*; *Voisin et al., 2009* |
| Lumican carryovers on neutrophils bound by Mac1 | Promote neutrophil migration via inducing outside-in signaling. | *Lee et al., 2009* |
| Vascular basement membrane/neutrophils | Deform passaging neutrophils and activate Piezo1 to promote bactericidal activity in tissue. | *Mukhopadhyay et al., 2024* |
| Collagen-IV/monocyte MMP9 | Digest locally the collagen-IV barrier to support trans-basement membrane migration of T cells. | *Watanabe et al., 2018* |

Extravasation journey stage:
Immediately after extravasation

| ECM interaction (protein or structure) | Function | Reference |
|---|---|---|
| Collagen/platelet GPVI | Partially activate platelets to adhere on and seal the exposed basement membrane at diapedesis sites with traffics overload. | *Gros et al., 2015*; *Currie et al., 2022* |
| Vascular basement membrane/endothelial focal adhesions | Enable endothelial force generation stimulated by platelet Ang1-Tie2 activation to build junctional stabilizing cortical actin bundles surrounding a passaging leukocyte. | *Braun et al., 2020*; *Braun et al., 2019* |

Extravasation journey stage:
in tissue

| | Function | Reference |
|---|---|---|
| Laminin-511/monocyte integrin α6β1 | Promote monocyte differentiation to macrophage. | *Li et al., 2020* |

*Table 1 continued*

**Extravasation journey stage:**
**Paracellular trans-endothelial diapedesis**

| | | |
|---|---|---|
| Collagen/MMP-8/9 and prolyl endopeptidase | Generate the matrikine PGP to secondarily chemoattract neutrophils, monocytes, and T cells. | *Gaggar et al., 2008*; *Pfister et al., 1995*; *Pfister et al., 1998*; *Weathington et al., 2006*; *Watanabe et al., 2018* |
| Laminin-γ2/neutrophil elastase | Generate the matrikine FGGPNCEHGAFSCPACYNQVKI to secondarily chemoattract neutrophils. | *Mydel et al., 2008* |
| Elastin/macrophage MMP-12 | Generate the matrikine VGVAPG to secondarily chemoattract monocytes. | *Senior et al., 1984*; *Taddese et al., 2009* |
| Laminin-511/leukocyte proteases | Potentially generate fragments containing the matrikine sequence AQARSAASKVKVSMKF to secondarily chemoattract neutrophils and monocytes. | *Adair-Kirk et al., 2003* |
| Versican/ADAMTS1 | Generate the matrikine (versikine) to modify macrophage cell state. | *Hope et al., 2016* |
| Versikine/TLR2 (and other unknown receptors) | Increase macrophage IL-1β and IL-6 productions and decrease IL-10 production. | *Hope et al., 2016* |
| | Increase macrophage IL-10 production in the presence of immune complex. | *Hope et al., 2016* |
| | Stimulate tumor cell production of monocyte and T cell chemokines. | *Hope et al., 2016*; *Hope et al., 2017* |
| Stiff tissue due to ECM build-up/leukocyte mechanosensors | Increase inflammatory cytokine production. | *Saitakis et al., 2017*; *Fahy et al., 2019*; *Chakraborty et al., 2021* |
| | Shift leukocyte migration from amoeboid to podosomal mode. | *Sridharan et al., 2019* |
| | Modulate phagocytosis. | *Sridharan et al., 2019*; *Adlerz et al., 2016*; *Mennens et al., 2017* |
| | Promote phagocyte proliferation. | *Chakraborty et al., 2021* |
| | Promote phagocyte glycolysis. | *Chakraborty et al., 2021* |
| | Sensitize phagocyte activation response to stimuli. | *Chakraborty et al., 2021* |
| | Promote efferocytosis by Piezo1 activation. | *Wang et al., 2024* |
| α2 Laminins and type-V collagen/(unknown monocyte Receptor) | Guide monocyte differentiation to efferocytic S1 macrophage by stimulating SHP-1 to suppress STAT-5 activity. | *Li et al., 2024* |

access of cellular or soluble blood components. The EC cytoskeleton links to the focal adhesion-ECM structure to form a mechanosensory structure containing proteins (such as talin and vinculin) capable of translating mechanical force into biochemical signals (*Finney et al., 2017*). This mechanosensing capacity, at least in part, organizes pericytes to perivascular space and subcellular junctional architecture of EC. Laminar shear stress in the blood flow stimulates endothelial abluminal secretion of PDGF-B to recruit tissue mesenchymal cells and differentiate them to pericytes (*Hsieh et al., 1991*; *Lindblom et al., 2003*; *Teichert et al., 2017*). These pericytes then release Ang1 to activate endothelial Tie2 signaling, strengthening endothelial junctions. This junctional strengthening signal is further potentiated by endocytosis of the VE-PTP, which inhibits Tie2, by laminar shear stress (*Shirakura et al., 2023*). Since both EC and pericytes contribute to ECM components in the basement membrane, this pericyte recruitment process may help formation of nascent basement membrane in angiogenic vessels (*Figure 1A*).

For a leukocyte to enter inflamed tissues, it engages in a series of steps well-described by the classical extravasation cascade (*Nourshargh and Alon, 2014*; *Vestweber, 2015*; *McEver, 2015*). After firm adhesion on the endothelium, down-tuning of integrin activity allows the leukocyte to crawl on the endothelium and search for a suitable tissue entry site (*Semmrich et al., 2005*; *Li et al., 2019*). At this stage, the continual endothelial monolayer and the underlying basement membrane are the sole barriers for leukocytes to overcome before tissue access (*Figure 1B*). The porous abluminal pericyte layer, in contrast, does not barricade leukocyte access. These barrier crossing processes, known as diapedesis, need to be tightly controlled to permit extravasation without disrupting general vascular structural integrity.

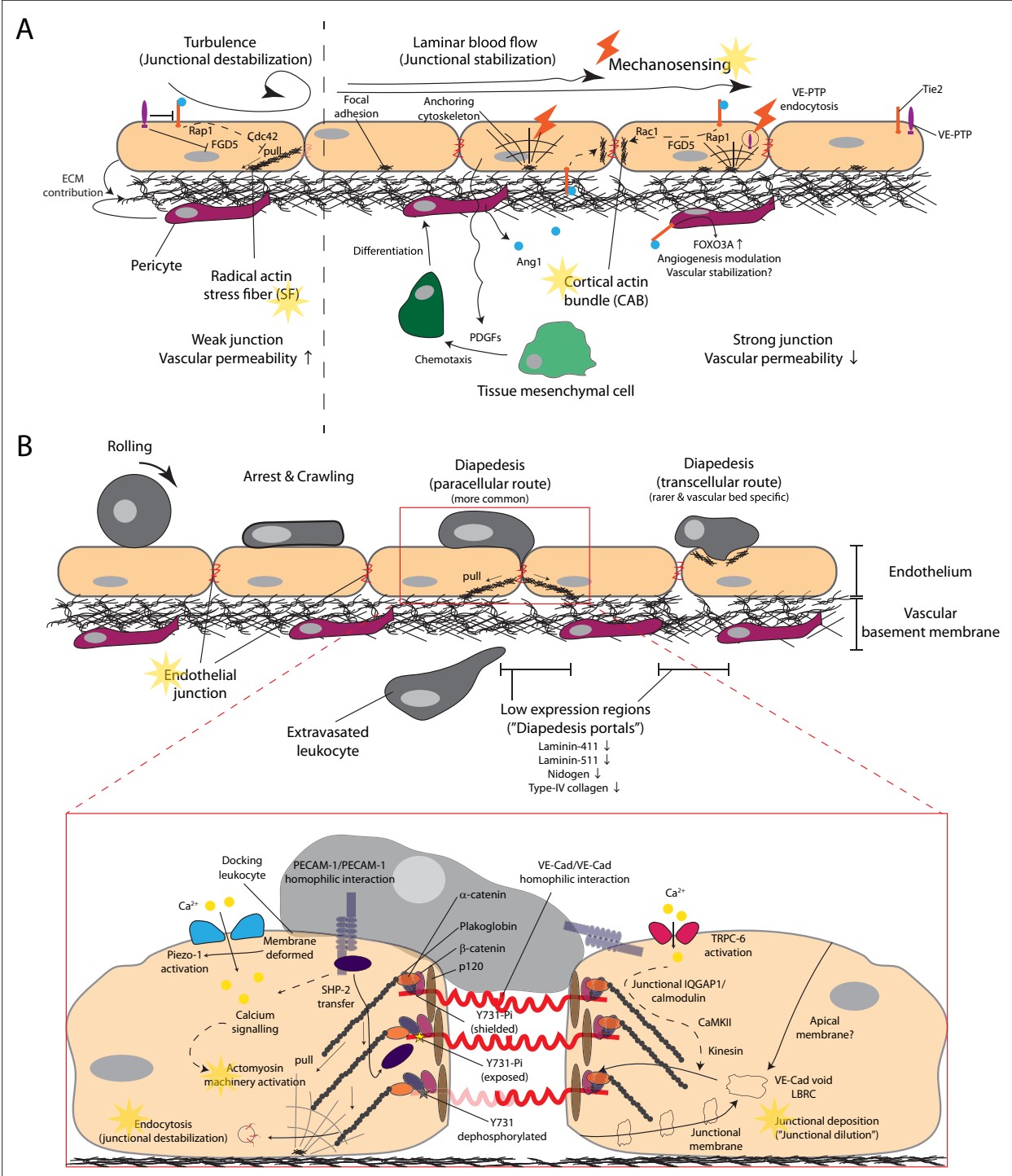

**Figure 1.** Vascular basement membrane enables endothelial mechanosensing and force-dependent junction remodeling to allow regulated leukocyte extravasation. (**A**) Vascular basement membrane supports endothelial mechanosensing via focal adhesion to construct leukocyte extravasation barriers by regulating junctional strength and recruiting pericytes. Tie2 signaling is activated by laminar blood flow to trigger force-dependent actin filament remodeling to strengthen junctions. (**B**) After establishing firm endothelial interaction, leukocytes prefer to paracellularly transmigrate through the vascular junctional barrier near the low expression regions on the vascular basement membrane. (inset) After leukocyte signals diapedesis initiation via calcium signaling, the basement membrane allows force generation in endothelial cells to locally remodel VE-Cadherin junctional barrier for leukocyte passage. Faint yellow stars, processes regulated by the vascular basement membrane as a collective structure. Dashed line, multi-step signaling processes.

## Paracellular versus transcellular route of entry across the endothelial barrier

A leukocyte could transmigrate across the endothelium either by diapedesis through the junctions (paracellular route) or directly through the EC (transcellular route) in vitro (*Carman and Springer, 2004*; *Ley et al., 2007*; *Nieminen et al., 2006*). This observation questions whether the endothelial junction or the apical surface represents the major endothelial barrier for leukocytes under physiological conditions. If one route can be specifically blocked, assessment of its influence on the extravasation response shall inform the importance of the blocked route.

VE-cadherin is a major regulator of the adherens junction and supports junctional integrity by homophilic dimerization of VE-cadherin from adjacent cells. Like other cadherins, VE-cadherin is bound at the C-terminus by intracellular β-catenin and α-catenin, which is then connected to the actomyosin contraction machinery (*Figure 1B*, inset). Since association to α-catenin stabilizes cadherin adhesion (*Ozawa and Kemler, 1998*), association or dissociation of catenin from VE-cadherin allows fine-tuning of junctional stability. In a previous study, transgenic mice with VE-cadherin genetically fused to α-catenin, thereby stabilizing the VE-cadherin junction and limiting paracellular diapedesis, were generated (*Schulte et al., 2011*). In these mice, the paracellular diapedesis events were reduced by ~50%, but the transcellular events were unaffected. In IL-1β inflamed cremaster, LPS stimulated lung, and a skin model of delayed type hypersensitivity, both neutrophil and lymphocyte extravasation were reduced by ~60%(*Schulte et al., 2011*). The similar reduction magnitudes of extravasation and paracellular diapedesis events suggest most of the reduced extravasation was due to blockade of paracellular diapedesis.

Although these observations suggest the endothelial junction is the sole barrier a leukocyte needs to overcome in vivo, 'closing' the paracellular route did not affect lymphocyte entry to lymph nodes via the HEV, indicating vessel- or tissue-specific preference of diapedesis route (*Schulte et al., 2011*). Later, it was demonstrated that the density of ICAM-1, an adhesion molecule supporting leukocyte adhesion on the endothelium, presented by the EC could influence the route decision, that a high ICAM-1 density favors the transcellular route for helper T cells across brain EC (*Abadier et al., 2015*). Hence, the preference for transcellular or paracellular route appears to depend on the adhesion strength between the leukocyte and the endothelium and the vascular bed in concern. Nevertheless, in many inflammatory conditions, the paracellular route of transendothelial migration remains the major route (*Schulte et al., 2011*). The intercellular adherens junctions, therefore, are the major barriers for leukocytes to cross the endothelium (*Figure 1B*).

## Vascular basement membrane supports force-dependent endothelial junctional remodeling required for leukocyte diapedesis across the endothelium

Actomyosin contraction in EC is required to support leukocyte paracellular transmigration (*Garcia et al., 1998*; *Saito et al., 2002*). The stable EC anchorage on the basement membrane via focal adhesions allows contractive force to be transmitted to actomyosin filaments at the cell periphery. It was hypothesized that such force pulling on endothelial junctional proteins helps open the junction for leukocyte passage, but molecular details had been obscure until recently.

Using a Förster resonance energy transfer (FRET) based molecular force sensor module inserted to the C-terminus of VE-cadherin, in which the FRET signal vanishes when the pulling force exceeds 4 pN, it has been elegantly demonstrated that endothelial mechanical force acting on VE-cadherin triggers a cascade of events that allow paracellular passage of a leukocyte (*Arif et al., 2021*). Multiple tyrosine residues, whose phosphorylation states control different endothelial functions, are present in VE-cadherin. Shown in vivo with knock-in point mutations, dephosphorylation of tyrosine at the Y731 position is required for leukocyte extravasation to inflamed tissues (*Wessel et al., 2014*). Upon leukocyte docking, EC-leukocyte PECAM-1 homophilic interaction triggers calcium signaling to activate the actomyosin machinery that pulls on VE-cadherin. The pulling force exposes the catenin-shielded Y731 site for dephosphorylation by the phosphatase SHP2 and induces endocytosis of VE-cadherin to loosen the junction for leukocyte passage (*Arif et al., 2021*; *Figure 1B*, inset). Since the force magnitude a cell generates correlates to the substrate rigidity (*Han et al., 2012*), endothelial cells on

a soft substrate cannot dephosphorylate Y731, showing force dependence of this process. Fluorescence lifetime imaging microscopy (FLIM) is a technique that measures the lifetime of a fluorophore, defined as the time the fluorophore stays excited, which depends on the fluorophore species and its surrounding microenvironment (*Datta et al., 2020*). Alteration in the FRET signal, due to the pulled VE-cadherin tension sensor, incurs a shift in fluorescence lifetime detectable by FLIM. FLIM images showed that only the local junction surrounding the passaging leukocyte was being pulled to not compromise the overall junctional integrity (*Arif et al., 2021*). Since not all junctions within the same EC were pulled, there involves a subcellular spatial restriction to confine the initiating calcium signal near the junction being pulled.

Such a subcellular cue might be provided by the leukocyte locally pressing the endothelial plasma membrane. A recent report describes the involvement of the tension-sensitive cation channel Piezo-1 in transendothelial diapedesis. Leukocyte docking changes the tension in the plasma membrane and provokes calcium influx via nearby Piezo-1 to activate the contractile actomyosin machinery (*Wang et al., 2022*; *Figure 1B*, inset). This model implies localized calcium signal, and thus the subsequent actomyosin contraction only occurs close to the leukocyte dock site. One would still need to explain why the leukocyte arrest site is not always the diapedesis site that a leukocyte usually crawls over a distance before beginning diapedesis (*Li et al., 2019*; *Goswami et al., 2017*).

There are other endothelial force-dependent processes supporting trans-endothelial diapedesis. Studies suggested directed membranous deposition, termed lateral border recycling compartments (LBRC), to the junction facilitates leukocyte passage. Deposition of LBRC to leukocyte-proximal junction requires force generation by the molecular motor kinesin along anchored microtubules (*Mamdouh et al., 2008*). Mechanistically, LBRC contains all other diapedesis-relevant junctional proteins but not VE-cadherin, and deposition to junctional site surrounding a transmigrating leukocyte dilutes the local VE-cadherin density and loosens the junction (*Mamdouh et al., 2003*). Regarding how LBRC spatially coordinates with the passaging leukocyte so that deposition is only directed to the leukocyte-contacted junction, again calcium signaling is involved. With calcium sensor mice that fluorescently visualize calcium signaling, leukocyte diapedesis triggers calcium signaling localized to the proximal junction. In this model, EC-leukocyte PECAM-1 homophilic interaction initiates calcium influx through the nearby calcium channel TRPC6. This activates calmodulin, immobilized at the junction via IQGAP1 binding, and downstream CaMKII to drive junctional deposition of LBRC (*Dalal et al., 2021*; *Figure 1B*, inset) TRPC6 can be activated by mechanical stress (*Brayden et al., 2008*), it might as well be activated by mechanical force exerted by leukocytes like it is the case for Piezo-1. While it is known that kinesin-microtubule machinery can be regulated by calcium signals (*Deavours et al., 1998*), the molecular details on its coupling to LBRC movement are unclear. Additional open questions, like whether and why LBRC membrane is exclusively from junctional membrane, and how VE-cadherin is excluded from entering this compartment, remain to be explored.

Overall, junction opening during paracellular transendothelial diapedesis involves numerous force-dependent processes and the basement membrane anchors endothelial cytoskeleton to allow force generation in EC.

## Pre-existing barrier weak spots with low laminin-α5 expression guide leukocyte diapedesis

Organized laminin structures known as the 'diapedesis portals' in the venular basement membrane directly signal endothelial junctional remodeling to regulate leukocyte diapedesis. Laminins are heterotrimers of the constituent chains (α-, β-, and γ- chains). Laminins α4β1γ1 and α5β1γ1 (laminin-411 and laminin-511) are the major laminins in the vascular basement membrane (*Timpl, 1996*). In vitro assays showed that laminin-511, but not laminin-411, activates RhoA/ROCK in EC to trigger junctional translocation of VE-cadherin, which stabilizes the junction and suppresses leukocyte transendothelial migration (*Song et al., 2017*). In the presence of shearing blood flow, laminin-511 also promotes p120 association to VE-cadherin and further stabilizes its presence at the junction (*Di Russo et al., 2017*; *Figure 2A*). These junction stabilizing properties of laminin-511 lead to the postulation that, if there are sites on the basement membrane with lower laminin-511 density, endothelial junctions near those sites would be more permissive to leukocyte passage. Such sites have actually been demonstrated.

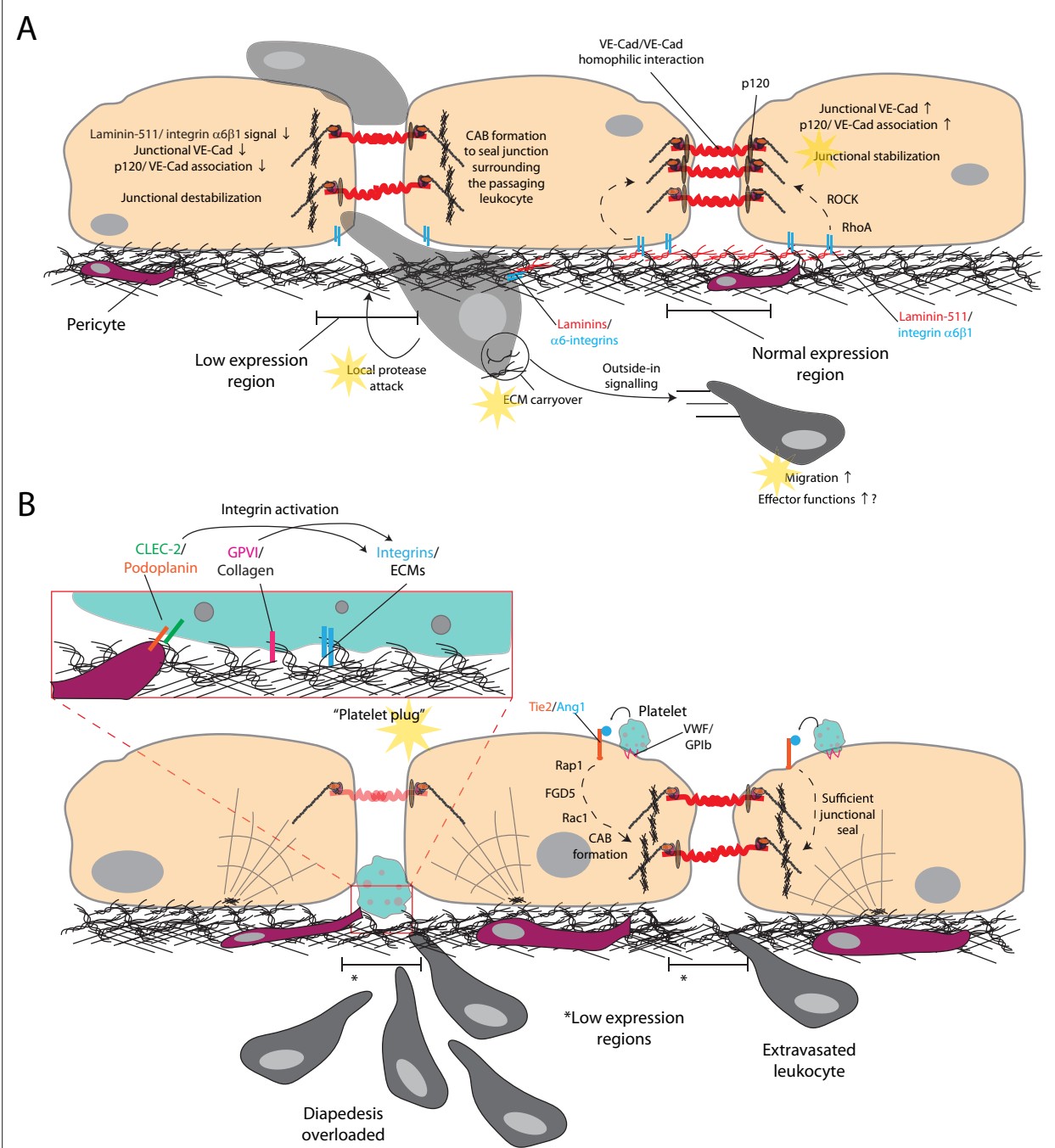

**Figure 2.** Regions in the vascular basement membrane with low expression of extracellular matrices are hotspots for leukocyte passage. (**A**) Laminin-511 in the vascular basement membrane signals endothelial cells to stabilize junctions via remodeling VE-Cadherin trafficking and interaction with associated intracellular stabilizing proteins. In the low expression regions, reduced local density of laminin-511, and thus the signaling to endothelial cells, loosens endothelial junctions to ease leukocyte diapedesis. (**B**) Typically, platelet-endothelium Ang1-Tie2 interaction keeps junctions surrounding the passaging leukocyte tight to prevent leak. However, over-usage of low expression regions exposes the basement membrane for partial platelet activation, generating a plug to prevent extravasation-associated vascular leak. Faint yellow stars, processes regulated by specific ECM proteins from vascular basement membrane. Dashed line, multi-step signaling processes.

Careful and detailed examination of the venular basement membrane ultrastructure by confocal microscopy with 3D reconstruction revealed that ECM component distribution is uneven and patchy with regions of low expression of laminin-511, laminin-411, nidogen, and type-IV collagen *Wang et al., 2006*; *Voisin et al., 2010*. The spatial analyses revealed the luminal sides of these low expression

regions are close to endothelial junctions, whereas the abluminal sides are typically not occupied by pericytes (*Wang et al., 2006*; *Figure 1B*). The reduced laminin-511 density at these regions locally 'loosens' the proximal luminal endothelial junctions for easier leukocyte access. Indeed, leukocyte diapedesis events were observed close to these low expression regions (*Wang et al., 2006*; *Sixt et al., 2001*). Trans-basement membrane diapedesis through these sites still requires α6 integrin to allow migration on laminins and protease activity to 'drill through' the basement membrane (*Dangerfield et al., 2002*; *Wang et al., 2005*; *Figure 2A*). In mice lacking laminin-α4, there is a compensatory upregulation of laminin-α5 resulting in a less patchy expression in the basement membrane. These mice consequently largely lack laminin-α5-low expression regions, and the resulting reduction in leukocyte extravasation confirms the facilitative role of the low expression regions (*Song et al., 2017*; *Wu et al., 2009*).

The exact mechanism how these low expression regions are formed in the first place remains elusive. One possibility is that these low-expression regions might be remnants of previous diapedesis events, which involve protease activities. As neutrophils pass through these low expression regions, these regions transiently enlarge with protease involvement including neutrophil elastase; the sizes recover to basal level over time (*Wang et al., 2006*; *Voisin et al., 2019*). The enlargement is at least in part related to deformability of the leukocyte since the passage of the more rigid neutrophils, but not the more flexible monocytes, resulted in low expression region enlargement (*Voisin et al., 2009*). In parallel, gaps between pericytes on the abluminal side of the basement membrane enlarge (*Proebstl et al., 2012*). An implication of these observations is that pericytes may have stretched these low expression regions in the basement membrane to enlarge them. Alternatively, the gap enlargement might be due to stretching by the rigid neutrophils or proteolytic activities followed by ECM redeposition. Direct visualization remains to be demonstrated, potentially by live confocal imaging and fluorescently labeled leukocytes, basement membrane, and pericytes. Whether and which component(s) of the basement membrane could be labeled by fluorescent proteins without disrupting its native organization remain to be investigated. It is unclear whether de novo deposition or the postulated basement membrane stretching by pericytes contributes to the recovery. If de novo deposition is involved, it remains to investigate how the deposition is controlled as not to bury and destroy these sites.

## How do leukocytes overcome barriers without vascular leak?

During paracellular transendothelial diapedesis, only the endothelial junction surrounding the passaging leukocyte is loosened. When asked if such localized junctional destabilization alone is sufficient to avoid vascular leak, it seems not to be the case. Complementary mechanisms must be involved, or the diapedesed leukocyte would leave a confined but leaky pore behind. This diapedesis pore must be sealed, either by endothelial junctional remodeling immediately behind the passing leukocyte or by a simple plug to prevent leak. This essential seal is provided by platelets. The vascular basement membrane coordinates these platelet processes through transient exposure of collagens or providing a structural support enabling force-dependent endothelial junctional remodeling.

In thrombocytopenic mice, which lack circulating platelets, skin inflammation induced by either UVB, croton oil, or immune complex (reverse passive Arthus reaction) caused bleeding at the inflamed site. This bleeding can be reduced by either leukocyte depletion or suppression of leukocyte-endothelial interaction via rolling/adhesion blockade or chemokine neutralization. Importantly, bleeding spots colocalized with extravasation sites, and specific blockade of leukocyte diapedesis with VE-Cadherin Y731F point mutation reduced bleeding. These suggest leukocyte diapedesis in the absence of platelets caused the bleeding (*Hillgruber et al., 2015*). Similar inflammatory bleeding could be observed in GPVI-/- mice whose platelets are present in blood but do not express GPVI (*Gros et al., 2015*). GPVI is an ITAM-containing receptor that signals via downstream Syk and Btk and activates platelets upon binding collagen, which is present in the vascular basement membrane or in tissue. Transfusion of untreated, but not Syk or Btk inhibited, wild-type platelets to GPVI-/- mice rescued the inflammatory bleeding, indicating GPVI/Syk/Btk signaling is required (*Gros et al., 2015*). These findings imply platelet contact with the collagenous vascular basement membrane signals and activates platelets via GPVI to plug the diapedesis pore left by the extravasated neutrophil. This has been recently directly visualized in a skin immune complex vasculitis model with real-time 4D intravital confocal microscopy that single platelets immediately adhered and plugged the blood vessel at the exact diapedesis site utilized by the passaging neutrophil. Examination of ultra-structures with transmission

electron microscopy found granules in the plugging platelets, suggesting these platelets had not been fully activated and degranulated (*Currie et al., 2022*). Consistently, a previous report showed platelet granules are not required for bleeding prevention in this inflammation model (*Deppermann et al., 2017*). Simultaneous antibody blockade of both GPVI and CLEC-2 resulted in much more bleeding, as well as more platelet leakage to extravascular space beyond the basement membrane, than GPVI single blockade (*Currie et al., 2022*). CLEC-2 binds podoplanin that is present on pericytes but normally not on EC nor in the basement membrane. Since both GPVI and CLEC-2 are activating receptors and platelet adhesion on the underlying basement membrane ECM likely involves integrins, GPVI, and CLEC-2 probably concert in transmitting inside-out signaling for integrin activation to support adhesive plugging (*Figure 2B*).

There is another mechanism where platelets prevent vascular leak via endothelial junctional remodeling. Inflamed ECs display von Willebrand factor (VWF) on the apical surface allowing platelet interaction (*Lowenstein et al., 2005*; *Bernardo et al., 2005*). VWF binding via GPIb activates platelets to luminally release angiopoietin-1 which activates Tie-2 (*Braun et al., 2020*; *Chow et al., 1992*; *Bryckaert et al., 2015*). Activated Tie-2 in EC phosphorylates downstream Rap1 and FGD5, which serves as a guanine exchange factor for Rac1 and Cdc42 to suppress stress fibers (SF) and to promote circumferential actin bundles (CAB; *Frye et al., 2015*; *Braun et al., 2019*). With a stable anchorage on the basement membrane such that tension of actin filaments could be exerted on junctions, CAB strengthens, whereas SF loosens endothelial junctions (*Figure 2B*). Endothelial conditional knockout or siRNA knockdown of Tie-2 phenocopied neutrophil diapedesis induced bleeding observed in thrombocytopenic mice (*Braun et al., 2020*). How this junction sealing process coordinates with leukocyte passage is unclear, or it just constantly seals any junctional pore incurred by leukocyte diapedesis.

If this Tie-2 centered mechanism is in place, why is platelet plugging necessary in immune complex-mediated inflammation to prevent diapedesis-mediated bleeding? It was found that platelet plugging occurred not at random diapedesis sites but was concentrated at hotspots experiencing on average 4 successive neutrophil diapedesis events (*Currie et al., 2022*). It thus appears that this Tie-2 mechanism is the default measure to seal the diapedesis pore, but may not catch up to the pace of leukocyte diapedesis in some inflammatory scenarios. On the other hand, leukocytes tend to take the partially breached, easier route for diapedesis. When the diapedesis breach overwhelms the Tie-2 mechanism to an extent that exposes the collagenous basement membrane, the platelet plug serves as the final defence line to prevent bleeding (*Figure 2B*).

Both Tie-2 and the downstream FGD5 are substrates of the inhibitory phosphatase vascular endothelial protein tyrosine phosphatase (VE-PTP). The Tie-2 mechanism could likely be strengthened by suppressing VE-PTP activity to counter vascular destabilization in inflammatory conditions with too rapid leukocyte extravasation. This approach has been shown to be beneficial in other vascular pathologies (reviewed in *Vestweber, 2021*). Alternatively, there have been proposals to use platelets as a drug delivery vehicle to tumors exploiting the thrombotic leaky tumoral vasculature (*Nishikawa et al., 2014*; *Li et al., 2016*). As both luminal VWF binding and the platelet plug involve partial platelet activation, similar platelet engineering approaches may be attempted to deliver inflammatory modulators in inflammatory diseases.

After overcoming the endothelial and basement membrane barriers, leukocytes reach the inflamed tissue to perform a multitude of functions. Instead of one-off barrier-crossing events, recent researches propose that previous endothelial and ECM interactions might impose lasting modulatory effects on leukocyte functions in tissue. In tissue, leukocytes are exposed to plenty of ECM components that interactions further affect leukocyte behaviors in tissue. All these dynamic events integrate to the decision on the tissue inflammation status: to sustain or to resolve the inflammation. A proper decision is crucial to avoid failed defence responses or detrimental chronic inflammation. The following sections will describe and discuss these pivotal aspects influencing the inflammation fate.

## Barrier passages modify leukocyte functions in tissue via biophysical stress, proteolysis, selective leukocyte passage, and spatial access to functional regulators

When a leukocyte diapedeses through the endothelial barrier, the cell body is heavily deformed with endothelial junctional proteins constantly tightening up the endothelial-leukocyte cell surface

interface to prevent leakage. Similar deformation is recorded when a leukocyte passages through the basement membrane barrier with the use of barrier-degrading proteases. These are biophysically and biochemically intense processes that might change the functional phenotype of the extravasated leukocyte. In addition, unique cellular interaction with EC or material interaction with the otherwise shielded basement membrane during extravasation may also modify leukocyte functions. Growing evidence has supported this view.

It has been recently shown that in neutrophils, 'squeezing' through confined space, corresponding to either the endothelial or basement membrane barrier in vivo, activates the tension sensor Piezo-1 to trigger calcium signaling and to upregulate expression of the cytotoxic hydrogen-peroxide-producing enzyme NOX4 (*Nisimoto et al., 2014*), resulting in increased bactericidal activity (*Mukhopadhyay et al., 2024*). Proteolytic activities involved during trans-basement membrane diapedesis of neutrophils resulted in carryover of laminin bits on cell surface after arriving at tissues (*Wang et al., 2006*; *Voisin et al., 2009*). While these bound laminins are bioactive as aforementioned, it is unclear whether they modulate neutrophil functions or interfere with crosstalk to other cell types. A potential hint comes from a similar carryover event with lumican, which is expressed by endothelial cells but not neutrophils. Lumican fragments were found on extravasated peritoneal neutrophils, but not on pre-extravasated blood or bone marrow neutrophils, and supported chemotactic migration and intra-tissue agility (*Lee et al., 2009*; *Figure 2A*). The enhanced motility likely involves lumican/Mac-1-mediated outside-in signaling. While outside-in signaling promotes leukocyte effector functions such as proliferation, cytokine production, myeloid degranulation, and phagocytosis, it remains to be formally tested whether leukocytes with lumican carryover show enhancement in these functions. Since laminin carryovers are also bound via integrins, modulation of outside-in signaling might represent a common mechanism of how ECM carryovers modify leukocyte functions. Another possibility is that lumican carryover might sensitize inflammatory responses as it bridges macrophage CD14 and bacterial LPS for better presentation to TLR4 (*Wu et al., 2007*).

For monocytes, any functional influence by extravasation may occur immediately at the level of the monocyte, during macrophage differentiation, or at the level of differentiated macrophage. A previous study showed that following trans-basement membrane diapedesis, some extravasated monocytes remained associated with the basement membrane where laminin-511 promotes transition of Ly6C$^+$MHC-II$^{lo}$ monocytes to Ly6C$^-$MHC-II$^{hi}$ macrophages (*Li et al., 2020*). While this function is consistent with the perivascular macrophages abundantly seen, it is unclear whether this differentiation boost is possible because passaging monocytes are in contact with the basement membrane in the first place, or monocytes have a natural tendency to migrate towards blood vessels. The latter scenario would argue this basement membrane function is standalone and does not require the extravasation process, that basement membrane laminin-511 would support monocyte differentiation even if monocytes are artificially injected into tissue. Either way, physiologically, exposure to laminin-511 in the basement membrane is only possible after extravasation.

Besides, monocyte/macrophage functions are affected by ontogeny (embryonic progenitor derived versus bone marrow derived) and lineage origins (*Lavine et al., 2014*; *Li et al., 2022a*). Previously described with inducible lineage tracing mice, the hematopoietic stem cells in bone marrow are partially derived from PDGFRα$^+$ mesenchymal progenitors transiently present during embryonic day E7.5-E8.5 (PDGFRα-lineage; *Ding et al., 2013*). As a result, about one-fourth of circulating monocytes is from the PDGFRα-lineage. Interestingly, vasculatures in skin and colon selectively prefer extravasation of PDGFRα-lineage Ly6C$^+$ monocytes. The responsible molecules nevertheless remain unclear. After differentiation to macrophages, PDGFRα-lineage cells show reduced DC-like characters and express a lower level of level of the inflammatory *Il1b* transcripts, suggesting less involvement in activating cellular engagement (*Li et al., 2022b*). Further mechanistic understanding of how PDGFRα-lineage macrophages are tilted to a less activating phenotype shall reveal new perspectives in inflammation control.

Collectively, functions of neutrophils and monocytes are modulated either directly via signaling events incurred by the extravasation processes, via selective extravasation of a leukocyte subset, or via novel access to function-modulating materials in tissues (*Figure 3*).

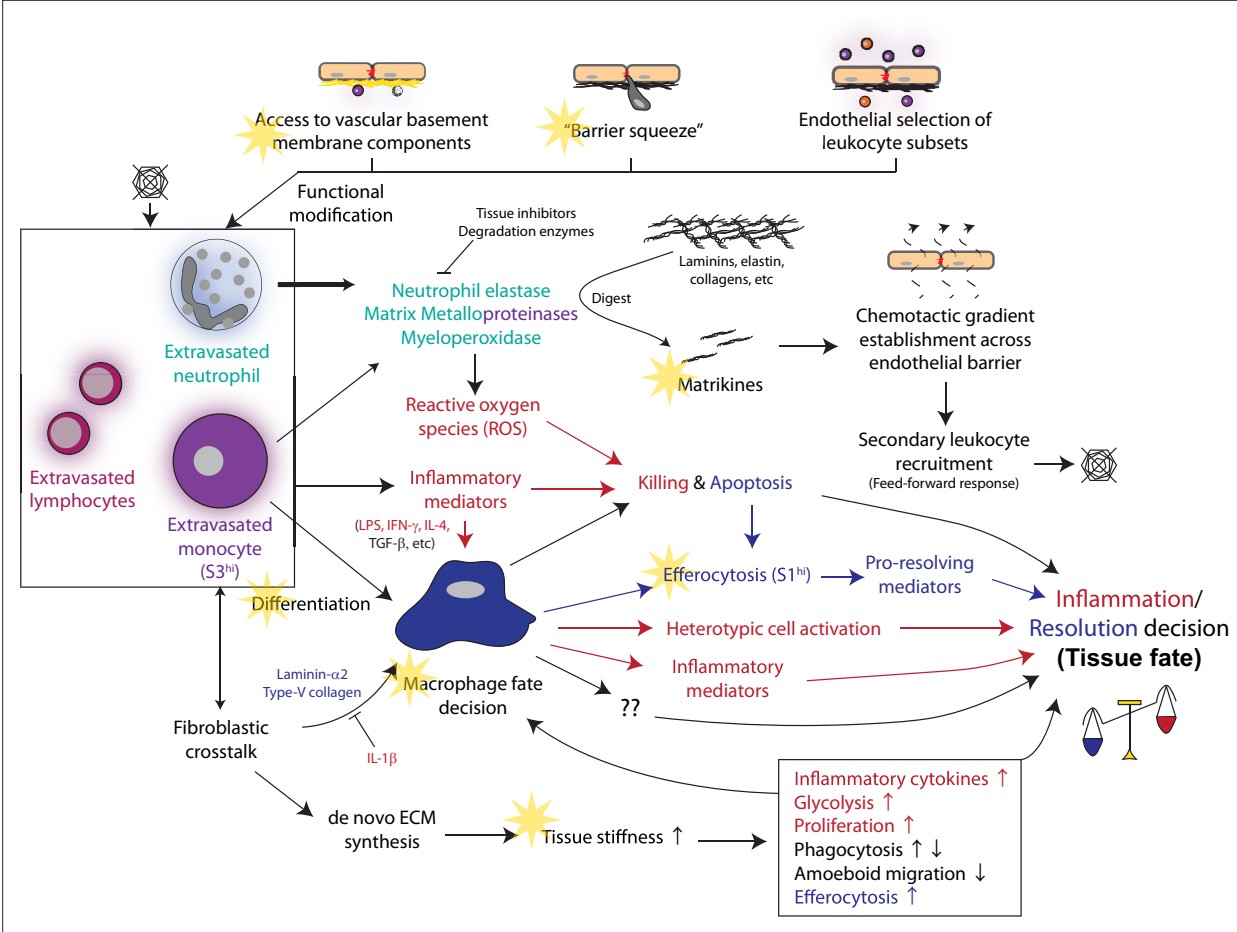

**Figure 3.** Extracellular matrices regulate various extravasated leukocyte activities in tissue and influence fate decision of the inflamed tissue. Biophysical and biochemical cues experienced during vascular barrier passage modify functions of extravasated leukocytes in tissue. Leukocytes also gain access to regulatory ECM components on the abluminal side of the vascular basement membrane as well as in tissue. Enzymes produced by leukocytes cleave their respective target ECM components to generate chemotactic matrikines leading to secondary leukocyte extravasation. Besides directly executing inflammatory functions, cytokines produced by leukocytes regulate ECM production by tissue stromal cells, which modulate tissue stiffness and leukocyte functions via mechanosensing. These dynamic leukocyte-ECM interactions and activities integrate to the tissue fate decision between sustained inflammation and resolution. Faint yellow stars, processes regulated by ECM proteins from vascular basement membrane or in tissue.

## Leukocyte proteases fragment tissue ECM to generate matrikines which promote secondary leukocyte trafficking

Proteases are powerful weapons utilized by innate leukocytes, especially neutrophils, to fight inflammation triggering invading pathogens after tissue entry. Extravasated neutrophils have experienced the 'squeeze' by the endothelial and basement membrane barriers and hence exhibit even stronger aggression towards pathogens. Delivery of neutrophil proteases stored in intracellular granules could take several forms. Besides the typical degranulation, neutrophils may choose to release neutrophil extracellular traps (NET) which comprise histone-bound chromatin, proteases, and additional offensive enzymes such as the reactive oxygen species (ROS) generating myeloperoxidase (*Brinkmann et al., 2004*). While a neutrophil cannot survive chromatin loss, in exchange, the sticky NET efficiently entangles pathogens for concentrated protease attack. Alternatively, proteases may be released as exosomes where individual molecules are oriented to resist antiprotease actions and to achieve stronger activities than free proteases (*Vargas et al., 2016*; *Genschmer et al., 2019*). However, protease activities not only destroy pathogens but also tissue ECM. Excessive protease activities could damage tissue integrity, as observed in chronic inflammation such as cystic fibrosis and chronic obstructive pulmonary disease (*Genschmer et al., 2019*).

Breakdown of tissue ECM produces a class of biologically active ECM degradation products termed matrikines that regulate leukocyte trafficking. These matrikines are often small enough to permeate the endothelial barrier to form a functional chemotactic gradient in vivo. Interestingly, a matrikine tends to promote chemotaxis of the leukocyte type releasing its synthesizing protease(s), constituting a feed-forward positive feedback loop (*Figure 3*).

Proline-Glycine-Proline (PGP) is a tripeptide derived from collagen under actions of neutrophil protease MMP-8/9 and prolyl endopeptidase (*Gaggar et al., 2008*). Acetylation at the N-terminus of PGP enables chemotactic activity for neutrophils via binding chemokine receptor CXCR1/2 (*Pfister et al., 1995*; *Pfister et al., 1998*; *Weathington et al., 2006*). This binding is explained by the structural similarity between PGP/CXCR2 and the binding of the cognate ligand KC to CXCR2 (*Weathington et al., 2006*). Leukotriene A4 hydrolase (LTA4H), which generates the well-known strongly chemotactic leukotriene B4, serves a surprising function to degrade PGP (*Gaggar and Weathington, 2016*). Another peptide corresponding to the 597–618 region of human laminin-γ2 chain was found to be chemotactic for neutrophils. Release of this peptide requires neutrophil elastase activity (*Mydel et al., 2008*). Proteolytic cleavage of elastin, abundantly found in elastic tissues such as skin and lung, by macrophage MMP-12 produces a hexapeptide VGVAPG to chemoattract monocytes via binding S-Gal receptor (*Senior et al., 1984*; *Taddese et al., 2009*). Laminin-511 in the vascular basement membrane potentially produces matrikines. A synthetic peptide (AQARSAASKVKVSMKF) derived from laminin-511 chemoattracts neutrophils and monocytes in vitro (*Adair-Kirk et al., 2003*). But it is unknown if this fragment is physiologically generated and if bioactivity requires exact sequence and length.

Besides myeloid leukocytes, in vivo, matrikines in tissue appear to be effective to chemoattract T cells as well. Although this implies targeting matrikine regulation could simultaneously regulate T cell traffic, the effect might be indirect. For example, in a vasculitis model where human arteries were implanted and inflamed subcutaneously to immunodeficient mice, systemic administration of PGP increased extravasation of both T cells and monocytes. However, monocyte or macrophage activity of MMP9 facilitates T cell migration, presumably by cleaving collagen-IV in the vascular basement membrane during extravasation (*Watanabe et al., 2018*). In human myeloma and colorectal cancer, tumors showing strong proteolytic activities on versican correlate to CD8$^+$ T cell infiltration. Versican, solely produced by macrophages, is cleaved by stromal cell-derived protease ADAMTS1 to generate a N-terminal matrikine fragment termed versikine (*Hope et al., 2016*; *Hope et al., 2017*). Further in vitro analysis, however, suggested that versikine alters macrophage cell state, which leads to production of T cell chemokines by tumor cells (*Hope et al., 2016*). Interestingly, intact versican and versikine appear to have inverse effects on inflammation since the intact form promotes anti-inflammatory IL-10 production while the fragment promotes inflammatory IL-1β production in macrophages (*Hope et al., 2016*; *Tang et al., 2015*).

Matrikines are regulated by intra-tissue availability of synthesis and degradation enzymes, as well as their regulators. For example, the presence of low-molecular weight hyaluronan stimulates tissue MMP-9 expression (*Fieber et al., 2004*) and may promote formation of PGP. Smoking strengthens macrophage expression of MMP-12 and suppresses the degradative LTA4H activity in lung, which could respectively promote accumulation of VGVAPG and PGP (*Woodruff et al., 2005*; *Noerager et al., 2015*). Bioactivity of MMPs, which are typically secreted as latent proproteins, requires activation. The zinc ion at the catalytic site, which is otherwise shielded by the thiol group of a cysteine residue of the pro-domain, needs to be exposed. Detailed activation mechanisms vary by the MMP species and are not fully known, but could involve cleavage of the pro-domain by furin or an active MMP or allosteric exposure of the catalytic site (*Springman et al., 1990*). On the other end, active MMPs could be inactivated by tissue inhibitors of metalloproteinases (TIMPs). Thus, matrikines contribute to secondary leukocyte recruitment caused by protease activities of extravasated leukocytes and further potentiate leukocyte activities in tissue. The multi-level regulation of matrikine ana-/catabolism implies their contribution to leukocyte activities in tissue likely varies by diseases; further investigations shall identify common features of conditions where managing matrikine-induced secondary leukocyte trafficking presents benefits against diseases.

## Tissue stiffness provides biophysical cues to modify extravasated leukocyte activities

On the contrary, instead of ECM degradation, there are disease conditions like skin keloid, bleomycin-induced pulmonary fibrosis, and liver fibrosis (cirrhosis) where inflammatory activities of extravasated

leukocytes promote amorphous collagenous deposition and harden the tissue. Keloid is considered a form of pathological scarring where initial wounding does not properly resolve but persists as chronic inflammation with undesirable accumulation of ECM. Without knowing the exact cause, multiple leukocyte types (including macrophages, mast cells, and regulatory T cells), growth factors and cytokines that promote fibroblast proliferation (TGF-β, PDGFs, FGFs) and angiogenesis (FGFs, VEGFs), and additional pleiotropic cytokines (IL6, IGF-1) contribute to this condition (*Lee et al., 2023*). Similarly, following an injury by excessive bleomycin, restless leukocyte activities in lung stimulate overgrowth of ECM-secreting fibroblasts. Cirrhosis could be caused by a variety of conditions, such as viral infection, prolonged alcohol overconsumption, bile flow obstruction, diets, and metabolic malfunctions, which lead to chronic hepatitis and fibroblast activation generating scar tissues in the liver. Further disease progression and functional decline could lead to liver failure and cancer. ECM build-up in these pathological scenarios alters tissue stiffness that translates to profound impacts on leukocyte functions (*Figure 3*).

Many leukocytes are mechanosensitive and change their behaviors according to the surrounding stiffness. Analogous to the EC-basement membrane binding described earlier, leukocytes bind the surrounding ECM substrates via focal adhesion to transmit the mechano-signals. In general, tissue hardening due to ECM accumulation drives inflammatory functions of leukocytes. Monocytes and T cells enhance their secretion of inflammatory cytokines in a stiff environment (*Saitakis et al., 2017*; *Fahy et al., 2019*). Most leukocytes migrate in amoeboid fashion in normal tissues. A study has shown that macrophages adopted this migration mode only on soft substrates (<88 kPa); on stiffer substrates (323 kPa), cells instead exhibited podosome-dependent migration (*Sridharan et al., 2019*). Many other leukocyte qualities relevant to inflammation, such as proliferation, glycolytic metabolism, and sensitivity to stimulus, are promoted by stiff substrates (*Chakraborty et al., 2021*). Phagocytic capacity may also be modulated by stiffness, but contradicting reports exist (*Sridharan et al., 2019*; *Adlerz et al., 2016*). One reason for this is that stiffness regulates the expression of specific phagocytic receptors and could differently affect phagocytosis towards different targets. Macrophage mannose receptor (CD206), which binds and supports endocytosis of ovalbumin, was downregulated on a stiffer substrate (12–50 kPa), when compared to a softer substrate (2 kPa). Accordingly, ovalbumin phagocytosis was reduced on the stiffer substrate, but transferrin endocytosis was unaffected (*Mennens et al., 2017*). Interestingly, this is an effect triggered only by medium stiffness since cells cultured on hard plastic (3 GPa) expressed more CD206. Such a non-linear effect on phagocytosis and the different ranges of substrate stiffness studied may explain the discrepancy between reports. In parallel, non-linear cellular responses to substrate stiffness have also been observed in regulatory T cells. Under induction conditions for regulatory T cells, regulatory T cell transition from helper T cells was more effective on a softer (<860 kPa) substrate than a stiffer (2600 kPa) one (*Shi et al., 2024*). However, in an even softer range of substrate stiffness (7.5–140 kPa), regulatory T cell formation was supported by the stiffer substrates via upregulating oxidative phosphorylation (*Shi et al., 2023*). Similarly, such non-linear stiffness responses have also been reported with helper and cytotoxic T cells (*Hickey et al., 2019*; *Yuan et al., 2021*). Of note, while stiffness responses by T cells require actomyosin cytoskeleton, partial inhibition of the actin polymerization, which slows remodeling or mechanotransduction, shifted the optimal stiffness. This optimal stiffness is also affected by the density of stimulatory ligands acting on T cells (*Yuan et al., 2021*). Empirically, tissue hardening often correlates to more severe inflammation. However, the non-linear stiffness responses described above imply that tissue stiffness might cooperate with many other response-modulatory factors to effect on leukocyte functions, and that its influence deserves to be examined in a disease-specific manner. In vivo studies of stiffness effects (e.g. hydrogel implantation), however, could be tricky without causing, or at least consideration of, structural disruption of the native tissue environment.

## Macrophage efferocytosis regulated by ECM is critical to timely subside inflammation and restore tissue homeostasis

To avoid these pathological events caused by abnormal and prolonged leukocyte activities, inflammation must be timely tamed as soon as the triggering pathogen is cleared. While pathogen removal reduces inflammatory reaction, many of the reactions triggered, such as the matrikine response and

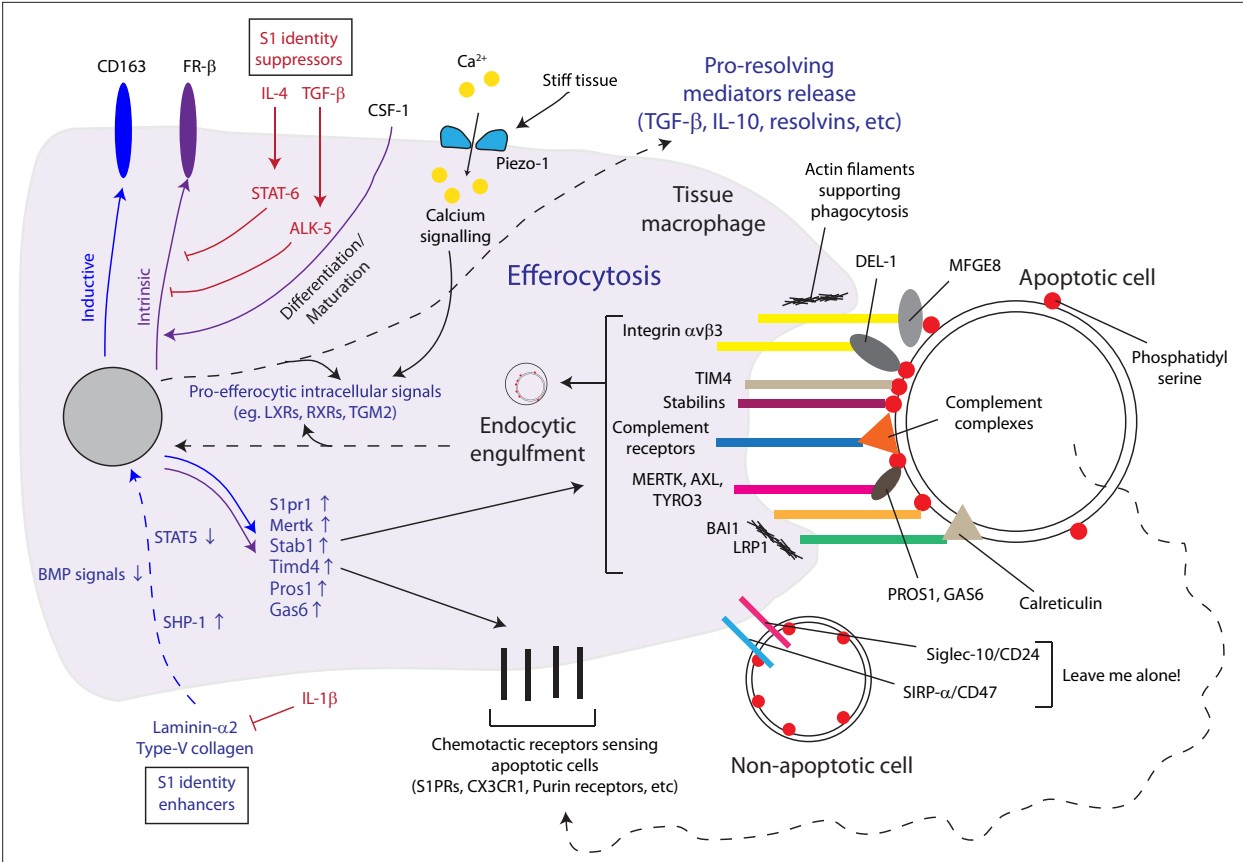

**Figure 4.** Efferocytosis exhibited by macrophages of strong S1 identity is an active driver of inflammation resolution and is regulated by specific extracellular matrices. Macrophages sense chemotactic materials released by apoptotic cells via an array of receptors to migrate towards and engulf them. Engulfment is mediated by receptor recognition of phosphatidylserine exposed on apoptotic cells either directly or through sandwiching adaptors. Non-apoptotic cells avoid engulfment via specific receptor complexes. Intracellular processing of endocytosed apoptotic cells activates efferocytic programs and releases anti-inflammatory mediators. Efferocytic macrophages show strong S1 identity, and the efferocytic capacity is under regulation by external signaling crosstalk with cytokines and extracellular matrices.

stiffness-driven inflammation, are self-sustaining. Active processes must be involved to stop inflammatory propagation.

Since neutrophils, which are the first to arrive at the inflamed tissue, have a relatively short lifespan (in days), a considerable number of apoptotic neutrophils has been accumulated by the time of pathogen clearance. Somatic cells damaged by inflammatory mediators add to the apoptotic counts. Macrophages are responsible for clearing up these dying cells to prevent necrotic release of intracellular materials that can trigger secondary tissue damages. Macrophages sense the chemotactic materials released by apoptotic cells and migrate towards them. Different from endocytosis of soluble and smaller particles, engulfment of apoptotic cells, either directly via receptor binding or indirectly via bridging adaptors, evokes intracellular signaling programs to acquire a pro-resolving cell state. This process, termed efferocytosis, produces anti-inflammatory mediators, such as IL-10 and resolvins, as end products (*Kourtzelis et al., 2019*; *Figure 4*) Efferocytosis thus serves as a switch for transiting inflammation to resolution. However, acquisition of the efferocytic phenotype is under competition of other fate guiding signals such as pathogen-derived LPS, IFN-γ, TGF-β, IL-4, and IL-1β, which promote killing responses, cross-activate other leukocytes like class-II helper T cells, or reduce tissue availability of efferocytic supportive signals (*Nathan et al., 1983*; *Stein et al., 1992*; *Mills et al., 2000*; *Figure 3*).

Efferocytosis could be facilitated with soluble factors produced by other cell types. Shown in ligation-induced periodontitis and monosodium urate crystal-induced peritonitis models, EC produces soluble DEL-1 that bridges phosphatidylserine exposed on apoptotic cells and integrin β3 on macrophages to ease recognition of target cells and facilitate efferocytic responses (*Kourtzelis et al., 2019*; *Choi et al., 2008*). Recently, fibroblastic ECM has been shown to guide monocyte differentiation

to macrophage of strong efferocytic capacity in a vitamin D3 analogue-induced atopic dermatitis model. In skin, monocytes mainly extravasate in hypodermis where laminin-α2 isoforms colocalize with type-V collagen. These ECMs activate the phosphatase SHP1 to suppress STAT5 activity during CSF-1-driven differentiation to form folate receptor β (FRβ)/CD163+S1 macrophage, with high expression of both chemotactic and efferocytic receptors towards apoptotic cells (*Li et al., 2024*). Previous studies showed that macrophages respond to BMP signaling (*Zhao et al., 2022*; *Ihle et al., 2024*) and endogenous BMP signals may also be involved in forming S1 macrophages. Like STAT5, suppressing BMP signals directly induced the S1 phenotype and prevented further induction by ECM. Boosting S1 differentiation with exogenous laminin-211 resolved atopic dermatitis; failure to form sufficient S1 macrophage, on the other hand, induced eosinophil maladaptation, tilting the dermatitis away from resolution.*Li et al., 2024* Intriguingly, while tissue hardening due to ECM build-up generally supports inflammation, a recent study showed that it also enhances macrophage efferocytosis via Piezo-1 activation, serving as an apparent countermeasure for the stiffness-induced inflammatory responses (*Wang et al., 2024*; *Figure 4*). Efferocytosis effectors have been proven beneficial in various experimental infections in the peritoneal cavity, liver, gastrointestinal tract, lung, and even the brain (*Dalli, 2017*). This process, therefore, presents an avenue for prospective clinical modulation of inflammatory diseases as reviewed in *Mehrotra and Ravichandran, 2022*.

However, efferocytic macrophages might not be protective in all scenarios. In the kidney, CD206+macrophages with transcript expression characteristics of the efferocytic S1 macrophages have been shown to cause renal fibrosis (*Cheung et al., 2022*; *Zimmerman et al., 2019*). In line, the S1-promoting niche laminin-α2 was found to accumulate in the fibrotic kidney in a genetic model of Alport disease (*Delimont et al., 2014*). The hardening fibrotic renal tissue may further promote efferocytosis via Piezo-1 activation as well. Taken together, while both pro-inflammatory and pro-resolving mediators co-exist in the inflamed tissue, accumulative cell apoptosis acts as a timed switch to provoke efferocytosis which gradually tips the balance towards the resolving side. While efferocytosis modulation has immense therapeutic potential, caution must be taken in a pathology-specific manner to evaluate its impacts on the disease.

## Prospects in inflammatory tissue fate modulation via navigating macrophage cell states

The central role of macrophage efferocytosis in determining tissue inflammation fate and the beneficial effects presented by efferocytosis effectors in numerous pathologies *Dalli, 2017* emphasizes vast therapeutic potential of this type of macrophage and tempts medicine development to boost its formation. Although we showed that type V collagen and α2-laminins could serve this purpose, *Li et al., 2024* their massive molecular structures hinder therapeutic uses. Rather, the downstream inactivation of STAT5 may be a more accessible target. Besides efferocytosis, this type of macrophage likely serves other functions, and whether these additional functions are beneficial, or at least unharmful, in all pathologies is debatable. It should be noted that pathological roles of macrophages have been frequently reported. Such a dual nature of macrophage cell state necessitates accurate control of macrophage cell state in pathology-specific manner. To effectively identify beneficial macrophage states with therapeutic potentials in various pathologies and to navigate other macrophage states to that direction, ideally, there is a coherent descriptive platform for macrophage cell states, each annotated with guiding regulators. Under such a unified platform, macrophage states responsible for producing various aforementioned ECM regulators that tip the fate decision of tissue inflammation could be clarified. Nevertheless, this apparently simple task is particularly challenging for the extreme phenotypical diversity macrophages present in vivo.

Since the last century, flexibility in macrophage response has been demonstrated in vitro: IFN-γ triggers a classical activation response, *Nathan et al., 1983* and IL-4 triggers an alternative response marked by mannose receptor upregulation (*Stein et al., 1992*). In 2000, the concept of M1/M2 first emerged to describe different in vivo macrophage tendencies using arginine to produce either cytotoxic nitric oxide (M1) or ornithine (M2) in different mouse strains. *Mills et al., 2000* Later studies associated M1 with the classical response, inflammation, and cytotoxicity, and M2 with the alternative response and anti-inflammation. Analyzing more in vitro stimulus-specific macrophage responses found the M1/M2 dichotomy over-simplistic, and M2 response was subdivided into M2a, M2b,

M2c, and M4 (*Mantovani et al., 2004*; *Gleissner et al., 2010*). It has now been recognized that this conventional M1/M2 response scheme is insufficient to capture in vivo macrophage responses (*Orecchioni et al., 2019*). Transcriptome-wide analysis of tissue macrophages in bulk attributed the complex macrophage states in vivo to their adaptive responses to the diverse tissue microenvironment (*Suzuki et al., 2014*; *Lavin et al., 2014*). In an effort to comprehend the complex in vivo macrophage cell states with common principles shared by all tissues, we performed a meta-analysis on published single-cell macrophage transcriptomes from various pathophysiological tissues and found macrophage populations shared between tissues in addition to reported tissue-specific macrophages. These shared macrophage populations could be segmented to five pan-tissue core identities (S1 to S5, defined by gene fingerprint). Under this framework, the macrophage transcriptomes and the identity kinetics during various pathologies were summarized in an archive termed Macrophage Identity Kinetics Archive (MIKA; *Li et al., 2024*). In order to cover more tissue types and conditions, we have expanded the archive to include 15 tissues with 20 pathophysiologies. During this expansion, we found additional pan-tissue cell states closely associated with the S1 identity. To address these S1 strays, we have replaced the original S4 identity, which has limited cell state defining efficiency, with two new core identities (new S4 and S6 identities). The current MIKA thus has six core identities in addition to tissue-specific macrophages (*Figure 5A*, Appendix 1 and *Supplementary files 1-3*).

Under the MIKA framework, different macrophage identities could be examined for regulatory roles on ECM functional effectors described in this review. Macrophages contribute minimally to vascular matrices, with nidogen-2 very weakly expressed by S1 macrophages. Beyond this expression level and amongst the many collagen isoforms, type-XIV and type-XXVII exhibit prominent expression by S4 identity and microglia, respectively, but functions of these rarer isoforms are less studied. Versican, which suppresses inflammation but generates the matrikine versikine upon proteolysis, is highly expressed by S3 identity. Instead of directly producing ECM, macrophages more often secrete fibroblastic growth factors, proteases, and protease inhibitors to regulate ECM kinetics in tissue. Specific macrophage identities expressing these regulators are outlined below. S1 and S4 identities secrete different isoforms of PDGFs. While TGF-β1 expression is detected in most identities, TGF-β2 expression is limited to cavity macrophages. Matrikine-generating MMP-8, MMP-9, and MMP-12 are expressed by S3, S1, and S4 identities, respectively. S1, S4, and several tissue-specific macrophages produce Serpin-B1 and -B6, which antagonize neutrophil proteases and may regulate matrikine production. Alveolar macrophages specifically express cathepsin K, which mediates phagocytic collagen catabolism and prevents lung fibrosis (*Fabrik et al., 2023*). For other proteases or protease inhibitors with fibrosis modulating capacity (MMP-13, MMP-14, ADAM-19, ADAM-33, PAI-2 [Serpinb2], Serpin-B8, Serpin-E2, and TIMP-2), contribution by specific macrophage identities could also be observed (*Figure 5B*). Overall, macrophages of S1, S3, and S4 identities as well as tissue-specific macrophages appear to be actively regulating ECM effectors with impacts on leukocyte functions. Of note, as S1 identity expresses PDGFs and MMP9, the strategy of promoting S1 macrophage formation to leverage efferocytosis for inflammation resolution shall be safe in conditions where these ECM regulators do not pose major concerns. In pathologies, macrophage identities dynamically fluctuate, as do their regulatory functions on the ECM functional effectors. Paying attention to these identity kinetics might hint at ECM malfunction, which radiates to leukocyte traffic and functions, in conditions where the ECM-regulating macrophage identities expand (*Supplementary file 1*). Efforts to develop medicines that offer freedom to modulate macrophage identity could thus tune ECM functions, in addition to other macrophage functions, and influence the inflammatory tissue fate.

## Concluding remarks

Extracellular matrix proteins are tightly coupled to the journey of a circulating leukocyte entering inflamed tissue. The vascular basement membrane anchors the endothelium to allow generation of cellular force required to remodel junctions for leukocyte passage. Leukocytes prefer using specialized diapedesis portals in the basement membrane, with weakened nearby endothelial junctions, to cross the vascular barriers. Optimal tissue entry requires aftercare of the 'diapedesis pore' left behind by the extravasated leukocyte, that is fulfilled by cooperative efforts of collagens in basement membrane, platelets, and EC. Both the vascular barrier passage and the myriad dynamic interactions with tissue cells and matrices influence leukocyte behaviors in tissue. These leukocyte activities, when properly orchestrated with macrophage fate, allow swift and timely restoration back to tissue homeostasis.

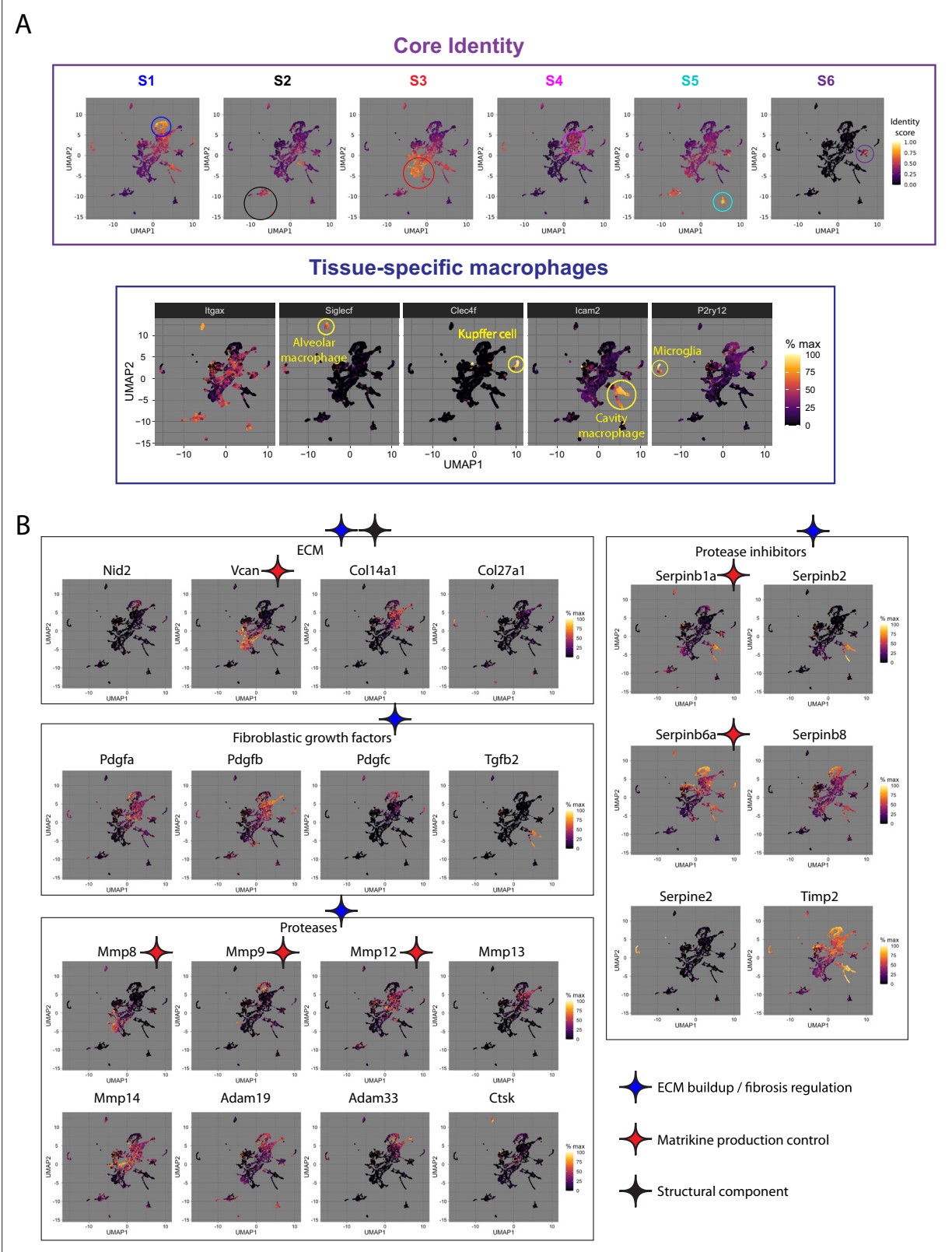

**Figure 5.** Extracellular matrix functional effectors are regulated by specific identities of tissue macrophage. (**A**) Under the pan-tissue macrophage identity framework MIKA, tissue macrophages are segmented to six core identities (**S1–S6**) and tissue-specific macrophages. (**B**) Macrophage identities expressing ECM structural components (black diamond), regulators of matrikine production (red diamond) or regulators of tissue ECM turnovers (blue diamond) are shown.

This extravasation journey of leukocytes illustrates the intermingled relationship among extracellular matrices, leukocyte extravasation, tissue activities of extravasated leukocytes, and the final tissue fate. We are still far from a complete understanding of this extreme complexity. Yet, persistent stepwise advances in our understanding shall offer plenty of regulatory targets for future modulation of inflammatory pathology.

## Additional information

### Funding

| Funder | Grant reference number | Author |
|---|---|---|
| Japan Society for the Promotion of Science | 25K19529 | Yu-Tung Li |

The funders had no role in study design, data collection and interpretation, or the decision to submit the work for publication.

### Author contributions
Yu-Tung Li, Conceptualization, Funding acquisition, Methodology, Writing – original draft, Writing – review and editing

### Author ORCIDs
Yu-Tung Li ⬛ https://orcid.org/0000-0002-0718-7344

Reviewer #1 (Public review): https://doi.org/10.7554/eLife.108284.3.sa1
Reviewer #2 (Public review): https://doi.org/10.7554/eLife.108284.3.sa2
Reviewer #3 (Public review): https://doi.org/10.7554/eLife.108284.3.sa3
Author response https://doi.org/10.7554/eLife.108284.3.sa4

## Additional files

### Supplementary files
Supplementary file 1. Macrophage identity kinetics of archived tissue conditions in MIKA. Sample-wise macrophage identity proportions, tissue origin, perturbation, GEO accession, and the definition of the input cell population for single-cell RNA sequencing were indicated. Identity proportions omitting monocyte-rich S3 were also provided, useful for datasets where baseline samples showed high S3 fraction. Note that the definition and experimental preparation of the input population affect subsequent efficiency of in silico purification of macrophages and the identity composition in a sample. This results in a discrepancy of baseline identity between datasets of the same tissue (eg. GSE180420 and GSE200115). Identity kinetics is best assessed with the same definition of the input population (usually within the same dataset).

Supplementary file 2. Gene correlation to each macrophage identity in MIKA. For each tissue (baseline or perturbed), Pearson correlation coefficients of each gene to each of the macrophage identity (S1 to S6) are tabulated. Moran I coefficient shows how focused the gene expression is on UMAP. For searches of identity markers, the correlation should be considered in adjunct with gene expression (*Supplementary file 3*); a gene of high correlation to an identity but with low expression likely underrepresents the identity of interest.

Supplementary file 3. Gene expression of each macrophage identity in MIKA. For each core macrophage identity (S1-S6) and tissue-specific macrophage (baseline or perturbated), average gene expression is tabulated in natural logarithmic scale.

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

## Appendix 1

## MIKA_v2

*Changes mentioned in this documentation serve for optimization and dataset expansion purposes, without affecting claims made in the original report (*Li et al., 2024*).

Changes from previous version:

1. Optimized pseudobulking algorithm
before:
Fixed pseudobulking factor calculated on per-sample basis to obtain 100 pseudobulks per sample. Each sample is independent when computed to pseudobulks.
current:
Pseudobulking factor is computed on a per-dataset basis (as an average of pseudobulking factors calculated on a per-sample basis in a dataset).
While maintaining the same target of 100 pseudobulks per sample and that seeds remaining to be chosen on a per-sample basis, all cells are candidates for pseudobulking.
Flexible pseudobulking factor is adopted to have more focal representation, such that the mean UMAP distance to a pseudobulking seed from member cells for pseudobulking is <0.05% of the mean UMAP distance to the seed from all cells.

2. Standardization of subtype scores
before:
After quantile-scaling gene signature expression across the cell population in a dataset, for each cell, expression of member genes in a subtype category is averaged to obtain a score for the subtype category. This score is quantile-scaled across the cell population to attain a standardized score ranging from 0 to 1 for each subtype category.
current:
The same procedures before standardization are followed. Scores for each subtype category are now standardized as a relative value to the cell population maximum, standardized score range remains from 0 to 1.

3. Criteria of "subtype qualifier" (for identity kinetics monitoring)
*This change is a follow-up to the changes in standardization of subtype scores.
before:
Standardized score of >0.7; for S3 omitted identity, S3 standardized score <0.5.
current:
Standardized score of >0.5; for S3 omitted identity, S3 standardized score <0.5.
Identity scores (i.e. quantitative measurement) are computed for each cell in each dataset; identity (i.e. categorical definition) is only assigned after pseudobulking and integration into MIKA based on the averaged identity scores and UMAP positions of the pseudobulks to enable identity kinetics monitoring without influence of tissue-specific macrophages (which often show compounded S1-S6 identity).

4. Integration of new datasets
before:
Include datasets covering 10 tissues and 12 conditions
current:
Include datasets covering 15 tissues and 20 conditions with the following datasets newly incorporated: (accession: tissue, perturbation, input cell population)

GSE275826: skin, aging, FACS-sorted skin cells of F4/80+MHCII+
GSE284253: aorta, abdominal aortic aneurysm, FACS-sorted CD45+
GSE230260: bone marrow, bone fracture, total cells
GSE241928: heart, myocardial infarction of different ages, bead-enriched CD45+

GSE290479: heart, anti-CD40 treatment, FACS-sorted CD45+
GSE269059: liver, diets, FACS-sorted CD45+
GSE274438: peritoneum, experimental endometriosis, FACS-sorted CD45+

*Identity kinetics of each dataset in MIKA is available in Supplementary Data 2.

5. Inclusion of new pan-tissue macrophage identities
before:
Five pan-tissue macrophage identities (S1-S5) are defined.
current:
Previous S4 identity is removed due to limited efficiency to define tissue-common macrophage cell states.

S4 and S6 (auxiliary) identities are newly defined to describe cell states straying from the S1 apex commonly found in multiple tissues. Current fingerprint to define six pan-tissue macrophage identities consists of 597 genes.

These updated pan-tissue macrophage identities and tissue-specific macrophage cell states are summarized in *Appendix 1—figure 1*.

## Glossary of selected terms

| Terms | Concise description |
|---|---|
| Diapedesis | The specific step(s) during leukocyte extravasation where a leukocyte transmigrates through a vascular barrier. May refer to trans-endothelial diapedesis or trans-basement membrane diapedesis. Different leukocyte and endothelial signaling events are involved depending on the barrier type and route (paracellular or transcellular) of diapedesis. |
| ECM carryover | After crossing the vascular basement membrane, fragments of the constituent ECM are bound on leukocytes as a result of proteolytic activities during the barrier passage. For some ECM, these fragments are reported to modulate leukocyte functions in tissue. |
| Efferocytosis | The process of macrophages (or phagocytes) engulfing apoptotic cells and thereafter synthesizing pro-resolving mediators. A process important to remove dead or dying cells accumulated throughout inflammation. Macrophages sense various cues released by apoptotic cells via a specific array of chemotactic receptors and migrate towards them. Surface exposure of phosphatidylserine on apoptotic cells triggers engulfment and subsequent pro-resolving signaling. |
| Extravasation | The entire multistep process of leukocyte crossing the vascular barriers and transmigrating from blood to tissue. Typically, a flowing leukocyte first tethers to the inflamed endothelium via catch-bond interaction with endothelial selectins. This initial tethering event is followed by decelerative rolling on endothelium and gradual activation of integrins on leukocytes. After firm arrest on the endothelium, the leukocyte crawls to search for a suitable site for trans-endothelial migration. Further trans-basement membrane migration and passage through any obstructing perivascular cells are required to gain access to the inflamed tissue. |
| Inflammatory tissue fate | An integrated decision that determines whether to resolve or to sustain an inflammation. This decision is dynamic and subject to the influence of myriad processes in the inflamed tissue, including the extravasation flux of leukocytes, leukocyte-ECM crosstalk during extravasation and in tissue, as well as other leukocyte and stromal cell activities. A correct decision at the right timing is crucial to promptly eliminate the initial cause of inflammation and to restore tissue homeostasis while preventing unnecessary tissue damage. |

*Continued on next page*

*Continued*

| Terms | Concise description |
| --- | --- |
| Matrikine | Bioactive peptide(s) produced from partial proteolysis of ECM. Different sets of proteases are responsible for digestion of different types of ECM. The resultant peptides are chemotactic to a specific leukocyte subset. |
| Mechanosensing | The process of a cell or cellular structure sensing the surrounding mechanical cues. This is achieved by binding to ECM present in the substrate via receptors (typically organized in focal adhesions) linked to the cytoskeleton decorated with mechanosensitive signaling proteins, which respond to and convert the mechanical cues to intracellular biochemical signals. |
| MIKA | A meta-transcriptome-based archive of tissue macrophage cell states and their kinetics in various pathophysiologies under a unified pan-tissue cell identity framework. Abbreviation of Macrophage Identity Kinetics Archive. The archive is expandable to accommodate new datasets. The current MIKA describes macrophages commonly found across multiple tissues and their kinetics with 6 core identities, S1 to S6, alongside tissue-specific macrophages. The archive classifies complex macrophage states in a tissue-borderless manner and serves as a common roadmap to navigate macrophage identity to pathology-resolving state(s) with identity regulators. |
| Secondary leukocyte extravasation | Additional leukocyte extravasation resulted from activities of extravasated leukocytes other than the initial cause(s) of the inflammation. This can be caused by de novo production of chemokines or matrikines (bioactive ECM fragments) from pre-existing ECM in tissue. |
| Tissue stiffness | The hardness of the tissue, or the elastic capability of the tissue to resist deformative force, usually expressed in Young's modulus. Tissue stiffness is affected by the amount, composition, and alignment of ECM in tissue. While an increase in tissue stiffness can be achieved by rapid ECM synthesis leading to tissue fibrosis, ECM usually has long turnover half-lives, and stiffness decline or fibrosis resolution is a rather gradual process without active intervention of proteases. Through receptor binding and mechanosensing, tissue stiffness can modulate intra-tissue leukocyte functions. |
| Vascular barriers | In general, the endothelium and the vascular basement membrane are the sole barriers for leukocyte extravasation. Depending on the route of diapedesis (paracellular or transcellular), either the endothelial junction or the cell body constitutes the endothelial barrier. |

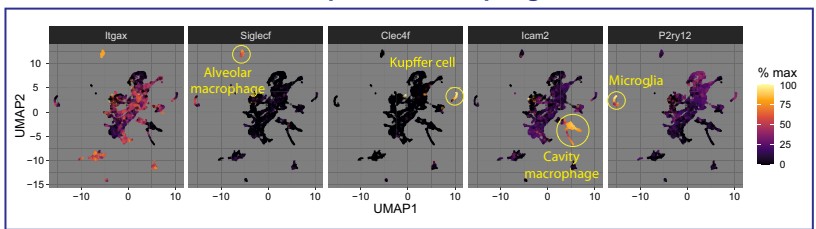

**Appendix 1—figure 1.** Updated identity definition in Macrophage Identity Kinetics Archive (MIKA). Expansion of the archive to include more pathophysiological conditions and tissues found S1-associated stray cell states (blue shade). The original S4 identity is replaced by two surrogate identities (new S4 and S6) to S1 identity to annotate the S1-strays (asterisks). Identity vector of each updated core identity is shown. Single hashtag indicates an ambiguous population with low gene expression. Tissue-specific states are marked by yellow circles or white arrows in tissue-expanded view.

