## [Editor Report · eLife Assessment]

This Review Article provides a timely review of how the extracellular matrix (ECM), particularly the vascular basement membrane, regulates leukocyte extravasation, migration, and downstream immune function, with a focus on monocytes/macrophages. It integrates molecular, mechanical, and spatial aspects of ECM biology in the context of inflammation, drawing from recent advances.

---

## [Referee Report · Reviewer #1 (Public review)]

Summary:

In this review, the author covered several aspects of the inflammation response, mainly focusing on the mechanisms controlling leukocyte extravasation and inflammation resolution.

Strengths:

This review is based on an impressive number of sources, trying to comprehensively present a very broad and complex topic. The revised version strengthens the connection with the ECM and all sections are now better integrated.

---

## [Referee Report · Reviewer #2 (Public review)]

Summary:

The manuscript is a timely and comprehensive review of how the extracellular matrix (ECM), particularly the vascular basement membrane, regulates leukocyte extravasation, migration, and downstream immune function. It integrates molecular, mechanical, and spatial aspects of ECM biology in the context of inflammation, drawing from recent advances. The framing of ECM as an active instructor of immune cell fate is a conceptual strength.

Strengths:

Comprehensive synthesis of ECM functions across leukocyte extravasation and post-transmigration activity.Incorporation of recent high-impact findings alongside classical literature.Conceptually novel framing of ECM as an active regulator of immune function.Effective integration of molecular, mechanical, and spatial perspectives.

Weaknesses:

Some sections remain dense with signalling detail.Figure readability could be improved through simplified labeling.

Appraisal and Impact:

The authors have achieved their aim of presenting an integrated view of ECM-immune interactions. The review provides conceptual and visual clarity on a complex topic.

---

## [Referee Report · Reviewer #3 (Public review)]

Summary & Strengths:

This review by Yu-Tung Li sheds new light on the processes involved in leukocyte extravasation, with a focus on the inter between leukocytes and the extracellular matrix. In doing so, it presents a fresh perspective on the topic of leukocyte extravasation, which has been extensively covered in numerous excellent reviews. Notably, the role of the extracellular matrix in leukocyte extravasation has received relatively little attention until recently. This review synthesizes the substantial knowledge accumulated over the past two decades in a novel and compelling manner.

The author discusses the relevant barriers leukocytes face during extravasation, addresses interactions with and transmigrate through endothelial junctions, mechanisms supporting extravasation, and how minimal plasma leakage is achieved during this process. The question whether extravasation affects leukocyte differentiation and properties is original and thought-provoking and has received limited consideration thus far. The consequences leukocytes extracellular matrix interaction, non-linear responses to substrate stiffness and effects on macrophage polarization, efferocytosis and the outcome of inflammation are relevant topics raised. Finally, a unifying descriptive framework MIKA is introduced, which provides a tool for classifying macrophages based on their expression patterns and could inform the development of targeted therapies aimed at modulating macrophage identity and improving outcomes in inflammatory scenarios.

In summary, this review provides a stimulating perspective on leukocyte extravasation in the context of extracellular matrix biology.

Weaknesses:

One potential drawback of this review is that the attempt to integrate a vast amount of information has resulted in complex figures, which may lead to important details being overlooked by readers.

---

## [Author Response]

The following is the authors’ response to the original reviews.

**Public Reviews:**

**Reviewer #1 (Public review):**
Summary:In this review, the author covered several aspects of the inflammation response, mainly focusing on the mechanisms controlling leukocyte extravasation and inflammation resolution.Strengths:This review is based on an impressive number of sources, trying to comprehensively present a very broad and complex topic.Weaknesses:(1) This reviewer feels that, despite the title, this review is quite broad and not centred on the role of the extracellular matrix.

Since this review focuses on the whole extravasation journey of leukocyte, this topic is definitely quite broad and covers several related fields. The article highlights the involvement of extracellular matrices (ECM), which are important regulators in multiple phases of the process, as a common theme to thread together these related topics. In the revised manuscript, we have made further emphasis on the role of specific ECM where appropriate (see point 2 below) and reorganized the last section to fit to this theme (see point 3 below).

(2) The review will benefit from a stronger focus on the specific roles of matrix components and dynamics, with more informative subheadings.

ECM may exert their roles either as a collective structure or as individual components. In the latter case, though the concerned ECM are specifically named throughout the manuscript, they may not be sufficiently obvious since they were often not mentioned in subheadings. For sections discussing functions of a specific ECM protein or at least a specific class of ECM proteins, we have now included their names in the subheadings as well for clarity (section 5 and 8). For other sections discussing functions that involve ECM as a macrostructure, either in form of vascular basement membrane to enable force generation or contributing to the overall tissue stiffness to provide biophysical cues (section 7, 9-10), we have included the specific processes regulated in the subheadings like that in section 4.

In the newly added discussion about the effects of matrikines on lymphocytes, we have also focused on the roles of specific ECM (PGP and versican; line 396-408). We hope these measures have made the subheadings more informative and provided better clarity of the roles of specific ECM components.

(3) The macrophage phenotype section doesn't seem well integrated with the rest of the review (and is not linked to the ECM).

Section 10-11 concerns how macrophage phenotypes affect the tissue fate following inflammation, that is, either to resolve inflammation and regenerate damages incurred or to sustain inflammation. This fate decision is an important aspect of this review: By furthering our understanding on the processes and mechanisms involved, we hope to gain the capability to properly control tissue outcomes in inflammatory diseases.

In section 10, an emphasis is put on macrophage efferocytosis, for its documented efficiency to resolve tissue inflammation. Specific ECM components (type-V collagens and 𝑎2-laminins) could directly promote macrophage efferocytosis (line 494-499). On the other hand, changes in tissue stiffness, as a result of ECM turnover regulated by activities of leukocytes or other cell types like fibroblasts as described in section 9, also affects efferocytosis (line 504-507).

We acknowledge that section 11 does not integrate well to the rest of the review, this section is now restructured. First, we describe how the ECM-regulated efferocytosis may be leveraged in disease modulation (line 522-529) and the need for a unified system to describe macrophage states for disease modulation (line 527-533) such that the responsible cell states for producing ECM regulators / effectors can be clarified (line 533-535). Given means to control macrophage cell states, this clarification will be useful to modulate pathologies involving ECM malfunctioning, that might be hinted by emergence or expansion of those responsible macrophage states in pathology (line 577-579, 581-585). Next, we provide historic background of efforts to establish such a unified descriptive platform for macrophage states (line 538-548) and describe the recent solution offered by MIKA. MIKA is a pan-tissue archive for tissue macrophage cell states based on meta-analysis of published single-macrophage transcriptomes, we have described the establishment, the latest development (Supplementary Data 1-4) and how the complex tissue macrophage states are segmented to core and tissue-specific identities under this framework (line 548-560, Figure 5A). Under this identity framework, expression of different ECM regulators discussed in this review (either the ECM per se, fibroblastic growth factors or proteases or protease inhibitors that regulate ECM turnover or matrikine production) are examined and linked to specific macrophage identities to offer insights of their potential relevance in pathologies (line 561-586, Figure 5B).

(4) Table 1 is difficult to follow. It could be reformatted to facilitate reading and understanding

We apologize for the complex setup. Table 1 is now reformatted to horizontal orientation to have enough space for the columns and reorganized for much easier comprehension.

(5) Figure 2 appears very complex and broad.

The original Figure 2 is now split to 2 separate figures (Figure 3-4). Since many processes of diverse natures influence tissue decision of resolution/inflammation, Figure 3 serves to outline and summarise these processes. Figure 4 now focuses on the regulation and tissue-resolving roles of macrophage efferocytosis, which specific ECM components (type-V collagens and α2-laminins) or tissue stiffness contribute to acquisition of this cell state. We hope this split can better focus the messages and ease understanding.

(6) Spelling and grammar should be thoroughly checked to improve the readability.

The manuscript is now proofread again, with corrections made throughout the text.

**Reviewer #2 (Public review):**
Summary:The manuscript is a timely and comprehensive review of how the extracellular matrix (ECM), particularly the vascular basement membrane, regulates leukocyte extravasation, migration, and downstream immune function. It integrates molecular, mechanical, and spatial aspects of ECM biology in the context of inflammation, drawing from recent advances. The framing of ECM as an active instructor of immune cell fate is a conceptual strength.Strengths:(1) Comprehensive synthesis of ECM functions across leukocyte extravasation and post-transmigration activity.(2) Incorporation of recent high-impact findings alongside classical literature.(3) Conceptually novel framing of ECM as an active regulator of immune function.(4) Effective integration of molecular, mechanical, and spatial perspectives.Weaknesses:(1) Insufficient narrative linkage between the vascular phase (Sections 2-6) and the in-tissue phase (Sections 7-10).

A transition paragraph between these two phases is now added between Section 6 and Section 7 to provide a narrative that ECM interaction events during extravasation affect downstream leukocyte functions (line 300-307).

(2) Underrepresentation of lymphocyte biology despite mention in early sections.

Although lymphocytes follow a similar extravasation principle as described in earlier sections, their in-tissue activities differ much from innate leukocytes. Discussion of crosstalk amongst T cells, innate leukocytes and matrikines is now incorporated into section 8 (line 396-408). Functional effects of tissue stiffness on different T cell subsets are now discussed in section 9 (line 456-469).

(3) The MIKA macrophage identity framework is only loosely tied to ECM mechanisms.

The involved section 11 is now restructured to better integrate to the ECM topics with the associated Figure 3 changed to Figure 5. Specifically, under the MIKA framework, we have now linked specific macrophage identities to expression / production of ECM functional effectors or regulators discussed in this review to highlight their regulatory roles and potential relevance in pathologies. Reviewer #1 and #3 also have raised this issue, please refer to the response to point (3) of reviewer #1 for detailed description.

(4) Limited discussion of translational implications and therapeutic strategies.

Besides translational implications or therapeutic strategies included in the original manuscript (line 291-298, 375-377, 421-424, 427-429, 508-511, 512-516 of the current manuscript), we have now included additional discussion to enrich these aspects (line 356-358, line 396-398, 402-403, 428, 436-439, 467-469, 523-536, 579-586).

(5) Overly dense figure insets and underdeveloped links between ECM carryover and downstream immune phenotypes.

The original Figure 1 containing the insets is now split to Figure 1-2 to avoid too dense information fitting to a single figure and to better focus the message in each figure. To resolve the issue of overly dense insets, insets in Figure 1 are redrawn/ reorganized. The original Figure 1C is moved to Figure 2A. The inset showing platelet plugging, together with the issue of diapedesis overloading described in the original Figure 1B, is reorganized to Figure 2B. In this way, Figure 1 focuses on the vascular barrier organization, overview of extravasation, and the force related events during endothelial junctional remodelling. Figure 2 focuses on the low expression regions, and junctional sealing processes after diapedesis.

We have now expanded discussion on ECM carryovers and their reported or implicated effects on downstream leukocyte functions (line 329-335).

(6) Acronyms and some mechanistic details may limit accessibility for a broader readership.

A glossary explaining specialized terms that may be confusing to readers of different fields is now included as Appendix 1 to broaden accessibility (line 977).

**Reviewer #3 (Public review):**
Summary & Strengths:This review by Yu-Tung Li sheds new light on the processes involved in leukocyte extravasation, with a focus on the interaction between leukocytes and the extracellular matrix. In doing so, it presents a fresh perspective on the topic of leukocyte extravasation, which has been extensively covered in numerous excellent reviews. Notably, the role of the extracellular matrix in leukocyte extravasation has received relatively little attention until recently, with a few exceptions, such as a study focusing on the central nervous system (J Inflamm 21, 53 (2024) doi.org/10.1186/s12950-024-00426-6) and another on transmigration hotspots (J Cell Sci (2025) 138 (11): jcs263862 doi.org/10.1242/jcs.263862). This review synthesizes the substantial knowledge accumulated over the past two decades in a novel and compelling manner.The author dedicates two sections to discussing the relevant barriers, namely, endothelial cell-cell junctions and the basement membrane. The following three paragraphs address how leukocytes interact with and transmigrate through endothelial junctions, the mechanisms supporting extravasation, and how minimal plasma leakage is achieved during this process. The subsequent question of whether the extravasation process affects leukocyte differentiation and properties is original and thought-provoking, having received limited consideration thus far. The consequences of the interaction between leukocytes and the extracellular matrix, particularly regarding efferocytosis, macrophage polarization, and the outcome of inflammation, are explored in the subsequent three chapters. The review concludes by examining tissue-specific states of macrophage identity.Weaknesses:Firstly, the first ten sections provide a comprehensive overview of the topic, presenting logical and well-formulated arguments that are easily accessible to a general audience. In stark contrast, the final section (Chapter 11) fails to connect coherently with the preceding review and is nearly incomprehensible without prior knowledge of the author's recent publication in Cell. Mol. Life Sci. CMLS 772 82, 14 (2024). This chapter requires significantly more background information for the general reader, including an introduction to the Macrophage Identity Kinetics Archive (MIKA), which is not even introduced in this review, its basis (meta-analysis of published scRNA-seq data), its significance (identification of major populations), and the reasons behind the revision of the proposed macrophage states and their further development.

The issue of section 11 being not well-integrated to the rest of the review has also been pointed out by other reviewers. In response, this section and the associated Figure 3 are now restructured for better integration to the theme of ECM. In brief, we have now discussed the regulatory roles of specific macrophage identities under the MIKA framework on the ECM regulators described in this review. Please refer to the response to point (3) of reviewer #1 for further details.

Regarding the difficulties in understanding the MIKA framework without prior knowledge of our previous work, first, we thank the reviewer for pointing out this issue and for making suggestion to better introduce the framework in a way easy to comprehend. Accordingly, in the current structure of section 11, we have described the rationales behind the needs of a common descriptive platform for tissue macrophage states (line 523-536), previous historic efforts (line 538-548), have introduced MIKA with mentions of the establishment and significance (line 548-555), and also have explained the rationales behind further development (line 555-560).

Secondly, while the attempt to integrate a vast amount of information into fewer figures is commendable, it results in figures that resemble a complex puzzle. The author may consider increasing the number of figures and providing additional, larger "zoom-in" panels, particularly for the topics of clot formation at transmigration hotspots and the interaction between ECM/ECM fragments and integrins. Specifically, the color coding (purple for leukocyte α6-integrins, blue for interacting laminins, also blue for EC α6 integrins, and red for interacting 5-1-1 laminins) is confusing, and the structures are small and difficult to recognize.

We apologize for the figures being too dense. Other reviewers have also raised this issue (see response to point (5) of reviewer #2 and response to point (5) of reviewer #1). The original Figure 1 and 2 are now reorganized to Figure 1-2 and 3-4 respectively, with insets also redrawn / expanded. Figure 1 now focuses on the vascular barrier organization, overview of extravasation, and the force related events during endothelial junctional remodelling. Figure 2 focuses on the low expression regions, and junctional sealing processes after diapedesis. Figure 3 serves to outline and summarise the diverse processes influencing tissue decision of resolution/inflammation. Figure 4 focuses on the regulation and tissue-resolving roles of macrophage efferocytosis. The original Figure 3, mainly concerning the methodological aspects of update of MIKA, is now integrated to Supplementary Data 1. This figure is now replaced as Figure 5 concerning the specific macrophage identities producing ECM effectors / regulators discussed in this review.

The concerned colour-coding issue is now in Figure 2A. All integrins are now in sky blue and all laminins in red. VE-Cad is also in red but has a different size and shape than laminins. We hope these modifications have improved the figures avoiding confusion.

**Recommendations for the authors:**
As you will see, the reviewers thought your manuscript was interesting and timely. However, as part 11 and its corresponding Figure 3 seem somewhat detached from the rest of the manuscript, one recommendation would be to remove this part for improved clarity. Other recommendations can be found in the comments below.
**Reviewer #2 (Recommendations for the authors):**
(1) Improve narrative linkage between vascular extravasation (Sections 2-6) and in-tissue leukocyte activities (Sections 7-10) by adding explicit transition text that connects ECM changes during transmigration to downstream immune cell phenotypes.

A transition paragraph is now added between section 6 and 7 (line 300-307).

(2) Expand discussion of lymphocyte-ECM interactions, either within existing sections or as a dedicated subsection.

We have now added discussion of the effects of matrikine on in vivo T cell traffic (line 396-409) and how T cell functions are regulated by tissue stiffness (line 457-466).

(3) Strengthen integration of the MIKA macrophage identity framework with ECM-specific drivers (e.g., stiffness, matrikines) and reduce methodological detail in Fig. 3 to focus on biological relevance.

We thank the reviewer for this recommendation and have adopted accordingly. First, the methodological details in the original Fig.3 is now integrated to Supplementary Data 1. This figure is now replaced as Fig.5 serving to examine different macrophage identities’ contribution to ECM effectors / regulators (specifically, ECM per se, growth factors for ECM-producing fibroblasts, proteases and protease inhibitors) discussed in earlier sections. Relevant texts are on line 561-586.

(4) Consider adding a glossary of key terms (e.g., matrikines, efferocytosis) to aid accessibility.

A glossary explaining selected terms that may be confusing to the general readership is now added as Appendix 1 (line 977).

**Reviewer #3 (Recommendations for the authors):**
The discussion of fibrosis as a significant consequence of inflammatory activity is currently limited to skin keloids and bleomycin-induced lung fibrosis. Considering the substantial clinical relevance, it would be beneficial to include a mention of the various forms of liver fibrosis resulting from chronic inflammation.

Liver cirrhosis is now mentioned as further examples of stiffening tissues on line 428, 436-439.

While the manuscript is generally well-written, there are several minor language issues that could be easily addressed by a native speaker during revisions. Some examples are listed below:

We thank the reviewer for these very helpful suggestions. They are adopted with the relevant line number in the revised manuscript indicated below. In addition, the manuscript is proofread again, with other grammatical mistakes corrected throughout the text.

(1) Line 40: ... proliferative pathogen, can be timely eliminated.

line 40

(2) Line 79: It may be worthwhile pointing out that while Claudin 5 expression is highest in the BBB, it is also relevant in the BRB and expressed at lower levels in peripheral ECs. Similarly, ZO-1 is widely found to be expressed in peripheral endothelial cells.

Thanks for indicating this caution, it is now mentioned on line 79-82.

(3) Line 82: affects leukocyte traffic and...

line 84

(4) Line 125: ..., both neutrophil and lymphocyte extravasation were reduced by ~60%

line 125-126

1. Line 128: The term "paracellular endothelial junction" is odd, as junctions are per se paracellular, i.e., between cells.

line 129

(6) Line 147: ... VE-Cadherin, in which the FRET signal vanishes.

line 148

(7) Line 186: "activation by direct leukocyte pressing" might be rephrased to be clearer, e.g. "it might as well be activated by mechanical force exerted by leukocytes like it is the case for Piezo-1."

line 185-186

(8) Line 216: The phrasing "knockout analogy" is somewhat unfortunate. I would suggest "...a4 ko mice consequently largely lack a5 low expression regions and the resulting reduction in leukocyte extravasation confirms the facilitating role of the low a5 expression regions."

line 217-218

(9) Line 219: ...how the low expression regions form / are formed in the first place... The term construction implies active planning.

line 220

(10) Line 278: ... thrombocytopenic mice ...

line 279

(11) Line 294: ... use platelets as a drug delivery vehicle ...

line 295

(12) Line 304: instead of "could have changed", use "might change"

line 315

(13) Line 320: at the level of the monocyte

line 336-337

(14) Line 324: ... consistent with ...

line 340

(15) Line 335: ... progenitors

line 351

(16) Line 432: ... a considerable number of apoptotic neutrophils has (been) accumulated

line 480

(17) Line 442: ..., which promote killing responses, cross activate other leukocytes ..., or reduce tissue availability...

line 490-491

(18) Line 453: ...This macrophage is responsive to BMP...

This sentence is now rephrased on line 500-501.

(19) Line 454: ...involved in forming S1 macrophages.

line 502

(20) Line 476: ...numerous pathologies...

Points (20-22) concerns Section 11, which is now restructured (line 523-586).

1. Line 492: ...macrophages acquiring phenotypes specific to their residence tissue.(22) Line 498: ...either - the tissue macrophage is of heterogeneous nature... or - tissue macrophages are of heterogeneous nature...